# Human embryo polarization requires PLC signaling to mediate trophectoderm specification

Meng Zhu[1,2], Marta Shahbazi[1,3†], Angel Martin[4†], Chuanxin Zhang[5,6], Berna Sozen[7,8], Mate Borsos[9], Rachel S Mandelbaum[10], Richard J Paulson[10], Matteo A Mole[1], Marga Esbert[11], Shiny Titus[11], Richard T Scott[11], Alison Campbell[12], Simon Fishel[12,13], Viviana Gradinaru[3], Han Zhao[6], Keliang Wu[5,6], Zi-Jiang Chen[5,6*], Emre Seli[11,14*], Maria J de los Santos[4], Magdalena Zernicka Goetz[1,7]

[1]Mammalian Embryo and Stem Cell Group, University of Cambridge, Department of Physiology, Development and Neuroscience, Cambridge, United Kingdom; [2]Blavatnik Institute, Harvard Medical School, Department of Genetics, Boston, United States; [3]MRC Laboratory of Molecular Biology. Francis Crick Avenue, Biomedical Campus., Cambridge, United Kingdom; [4]IVIRMA Valencia, IVI Foundation, Valencia, Spain; [5]Center for Reproductive Medicine, Cheeloo College of Medicine, Shandong University, Jinan, China; [6]Key laboratory of Reproductive Endocrinology of Ministry of Education, Shandong University, Jinan, China; [7]Developmental Plasticity and Self-Organization Group, California Institute of Technology, Division of Biology and Biological Engineering, Pasadena, United States; [8]Yale School of Medicine, Department of Genetics, New Haven, CT, United States; [9]California Institute of Technology, Division of Biology and Biological Engineering,, Pasadena, United States; [10]USC Fertility, University of Southern California, Keck School of Medicine, Los Angeles, United Kingdom; [11]IVIRMA New Jersey, Basking Ridge, NJ, United States; [12]CARE Fertility Group, John Webster House, 6 Lawrence Drive, Nottingham Business Park, Nottingham, United Kingdom; [13]School of Pharmacy and Biomolecular Sciences, Liverpool John Moores University, Liverpool, United Kingdom; [14]Yale School of Medicine, Department of Obstetrics, Gynecology, and Reproductive Sciences, New Haven, CT, United States

**\*For correspondence:**
Chenzijiang@vip.163.com (Z-JC);
Emre.Seli@yale.edu; emre.seli@ivirma.com (ES)

†These authors contributed equally to this work

**Competing interest:** The authors declare that no competing interests exist.

**Abstract** Apico-basal polarization of cells within the embryo is critical for the segregation of distinct lineages during mammalian development. Polarized cells become the trophectoderm (TE), which forms the placenta, and apolar cells become the inner cell mass (ICM), the founding population of the fetus. The cellular and molecular mechanisms leading to polarization of the human embryo and its timing during embryogenesis have remained unknown. Here, we show that human embryo polarization occurs in two steps: it begins with the apical enrichment of F-actin and is followed by the apical accumulation of the PAR complex. This two-step polarization process leads to the formation of an apical domain at the 8–16 cell stage. Using RNA interference, we show that apical domain formation requires Phospholipase C (PLC) signaling, specifically the enzymes PLCB1 and PLCE1, from the eight-cell stage onwards. Finally, we show that although expression of the critical TE differentiation marker GATA3 can be initiated independently of embryo polarization, downregulation of PLCB1 and PLCE1 decreases GATA3 expression through a reduction in the number of polarized cells. Therefore, apical domain formation reinforces a TE fate. The results we present here demonstrate how polarization is triggered to regulate the first lineage segregation in human embryos.

## Introduction

Mammalian development begins with a series of cleavage divisions of the zygote during which cells progressively differentiate from each other to establish embryonic and extra-embryonic tissues, which will be essential for embryo implantation and further development. Three lineages have to become established by the time of embryo implantation: the trophectoderm (TE), which will generate the placenta, the epiblast (EPI), which will form the embryo proper, and the hypoblast, which will form the yolk sac (*Shahbazi and Zernicka-Goetz, 2018*).

Studies in the mouse embryo showed that the physical process of lineage segregation begins upon embryo compaction and polarization at the late eight-cell stage, the beginning of day 3 of development. The processes of embryo compaction and polarization lead to the closer apposition of neighboring blastomeres and the formation of an apical domain that orients towards the outside of the embryo (*Cockburn and Rossant, 2010*; *Fleming and Johnson, 1988*; *White et al., 2016*). The differential inheritance of the apical domain during the successive cleavage divisions generates polarized and non-polarized cells that sort to the outside and inside of the embryo, respectively (*Anani et al., 2014*; *Johnson and Ziomek, 1981*; *Samarage et al., 2015*). Subsequently, cell polarity and cell position will regulate the expression of lineage-specific transcription factors, and consequently, the outer polarized cells will differentiate into TE, and inner non-polarized cells will retain pluripotency and become the inner cell mass (ICM), which will give rise to the EPI and hypoblast lineages (*Chazaud and Yamanaka, 2016*; *Zhu and Zernicka-Goetz, 2020*).

Previous studies indicated that compaction of human embryos takes place between days 3 and 4 after fertilization (*Iwata et al., 2014*; *Nikas et al., 1996*). However, the underlying mechanism of embryo polarization and regulation of its timing has remained unknown. Here, we address these questions and show that apical domain formation occurs in two steps – an initial apical enrichment of F-actin, followed by the polarization of the PAR complex. Mechanistically, the establishment of the apical domain requires Phospholipase C (PLC) signaling, and interfering with this process leads to a decrease in the levels of the TE marker GATA3.

## Results

### Dynamics of cell polarization in the human embryo

To determine the timing of cell polarization in the human embryo, we carried out detailed analyses on a total of 260 supernumerary in vitro fertilized human embryos that were cryopreserved at embryonic day 3. Embryo development was recorded using time-lapse imaging until embryonic day 4 when embryos were fixed and analyzed by immunofluorescence (*Figure 1 a* ). At the start of the experiment, embryos had a mean number of eight cells that increased to approximately 12 cells after 24 hr in culture (*Figure 1—figure supplement 1a-d*). Our analyses revealed that the cell cycle time decreased from the 4–8 cell stage transition (22.15 hours) to the 8–16 cell stage transition (14.49 hr) (*Figure 1—figure supplement 1e*). The majority of analyzed embryos (89.4%, N = 132) showed no signs of degenerated cells, both after warming (89.4%, N = 132; *Figure 1—figure supplement 1f*) and after 24 hr of culture (95.3%, N = 85; *Figure 1—figure supplement 1g*), indicating embryo quality was not compromised during the experiments.

To quantify accurately the dynamics of embryo compaction, we measured the inter-blastomere angle between adjacent blastomeres as embryos developed in culture. Based on previously established criteria (*Zhu et al., 2017*), we defined the initiation of compaction as the time at which the inter-blastomere angle increased to greater than 120° between any two blastomeres, and the completion of compaction as the time at which all angles were greater than 120°. In this way, we were able to reveal that the average cell number in the embryo at the start of compaction was 10.14 (N = 83), increasing to 11.57 (N = 54) by completion of compaction (*Figure 1—figure supplement 1h*).

In addition, we analyzed embryos in which the exact times of fertilization and cryopreservation were known. These analyses revealed that the beginning of compaction was accomplished on average 78.2 ± 6.6 hours post-fertilization, while the completion of compaction was accomplished at 80.2 ± 8.5 hr (*Figure 1 b – d* ; *Figure 1—figure supplement 1i*). These observations indicate that the onset of compaction in human embryos is heterogeneous, and while compaction starts at the eight-cell stage, it extends to the 8–16 cell stage transition. This finding is consistent with previous observations on human embryos (*Iwata et al., 2014*; *Nikas et al., 1996*; *Skiadas et al., 2006*) and contrasts with

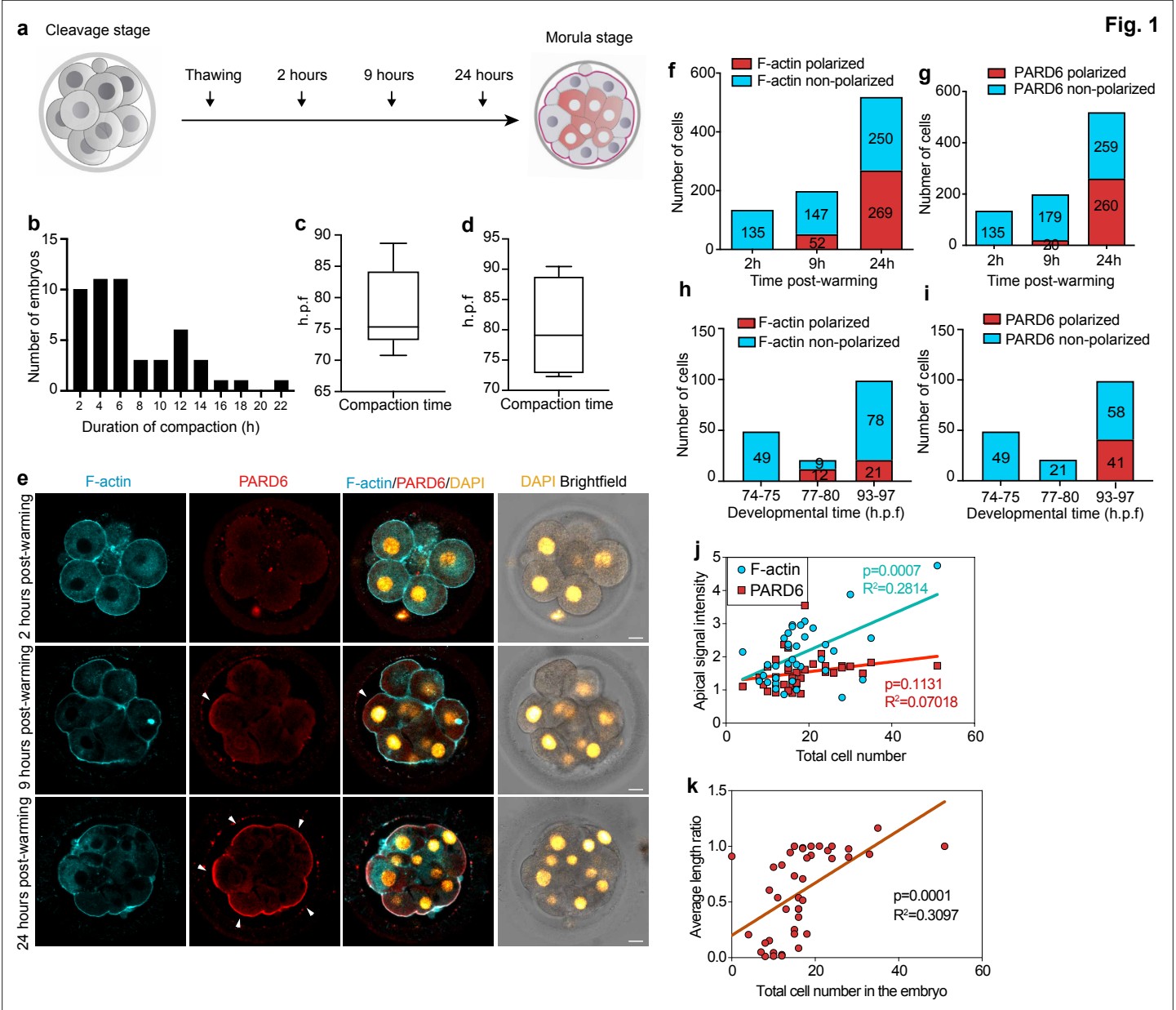

**Figure 1.** Timing of compaction and polarization in human embryos. (**a**) Scheme for human embryo culture. Supernumerary in vitro fertilized human embryos were warmed at day 3, and cultured for 2, 9, or 24 hr to examine the localization of polarization markers. (**b**) Histogram showing the time between the initiation and the end of compaction. N = 50 control embryos. Eight independent experiments. (**c**) Time at which compaction is initiated in embryos with known fertilization time. N = 10 control embryos. (**d**) Time at which compaction is completed in embryos with known fertilization time. N = 4 control embryos. Data is shown as a box and whiskers plot from the minimum to the maximum value. Four independent experiments. (**e**) Representative images of human embryos fixed at different developmental time-points (as shown in a) and immunostained for F-actin and PARD6. Arrowheads indicate the apical domain. (**f**) Quantification of the number of cells showing polarized or non-polarized F-actin in different developmental time-points. (**g**) Quantification of the number of cells showing polarized or non-polarized PARD6. For both e and f, the number in each bar represents the number of cells analyzed. N = 17 embryos for 2 hr post-warming; N = 20 embryos for 9 hr post-warming and N = 32 embryos for 24 hr post-warming. N = 5 independent experiments. (**h**) Quantification of the number of cells showing polarized or non-polarized F-actin in different post-fertilization time points. (**h**) Quantification of the number of cells showing polarized or non-polarized PARD6 in different post-fertilization time points. For both g and h, the number in each bar represents the number of cells analyzed. N = 6 74–75 h.p.f., N = 2 77–78 h.p.f. and N = 9 h.p.f. embryos. (**i**) Correlation between F-actin/PARD6 apical enrichment and cell numbers. Apical enrichment of F-actin and PARD6 is measured as the ratio of signal intensity on the apical surface to the cell-cell contacts. (**j**) Correlation between the length of the apical domain (based on the PARD6 immunostaining) and embryo cell numbers. Each dot represents one analyzed cell. h.p.f.: hours post-fertilization. Seven independent experiments. Scale bars, 15 µm.

The online version of this article includes the following figure supplement(s) for figure 1:

*Figure 1 continued on next page*

*Figure 1 continued*

**Source data 1.** Source data for *Figure 1*.

**Figure supplement 1.** Morpho-kinetic analysis of human embryos cultured in vitro from day 3 to day 4.

observations on mouse embryos where compaction is completed by the end of the eight-cell stage (*Levy et al., 1986*; *Zhu et al., 2017*).

We next wished to determine the sequence of developmental events of human embryo polarization in relation to compaction. In the mouse embryo, the localization of the PAR complex to the apical domain serves as the crucial developmental transition that controls embryo polarization and the expression of TE lineage-specific transcription factors (*Sasaki, 2015*). Transcriptomic analyses of human embryos indicate that PAR complex components, such as PARD6, are also expressed in human embryos at the eight-cell stage (*Petropoulos et al., 2016*; *Stirparo et al., 2018*). We, therefore, examined the spatial localization of PARD6 and F-actin in embryos at three different time points during their in vitro culture (*Figure 1 a* ). We found that none of the embryos were compacted after 2 hr of culture, in agreement with our time-lapse studies above (N = 18 embryos). At this stage, both cortical F-actin and PARD6 were uniformly distributed within the embryos, indicating that they had not yet initiated polarization (N = 135 blastomeres; 17 embryos) (*Figure 1 e–g*). However, 7 hr later, we found that 26 % of embryos had become fully compacted (N = 199 blastomeres; 19 embryos). All compacted blastomeres displayed an apical enrichment of cortical F-actin at the cell-contact free surface (*Figure 1e –g*). In contrast, only half of these compacted blastomeres showed PARD6 polarization (26.3%, N = 199 blastomeres; 20 embryos) (*Figure 1e –g*). The proportion of compacted embryos in which both F-actin and PARD6 were polarized increased significantly during the next 15 hr of culture (91.0 % for F-actin and 71.5 % for PARD6 polarization; 32 embryos) (*Figure 1e –g*). Quantification of polarization in embryos in which the exact times of fertilization and cryopreservation were known revealed a similar pattern. Specifically, none of the embryos showed F-actin nor PARD6 polarization (N = 49 blastomeres; six embryos) at 74–75 hours post-fertilization. We were able to detect F-actin polarization at 77–80 hours post-fertilization (57.1 %; N = 21 blastomeres; two embryos), and PARD6 at 93–97 hours post-fertilization (41.4 %; N = 99 blastomeres; nine embryos) (*Figure 1h–i*). The level of PARD6 apical enrichment, the size of the PARD6 domain, and the number of blastomeres showing PARD6 polarization were positively correlated with embryo cell numbers, indicating that the apical localization of PARD6 increases as embryos develop in culture (*Figure 1j –k* and *Figure 1—figure supplement 1j*).

To determine whether other components of the apical PAR polarity complex also become localized to the apical domain at the same time, we examined the distribution of the atypical Protein Kinase C (aPKC). We found that aPKC localized to the cell-contact free surface at the same developmental time frame (N = 62 blastomeres; four embryos) (*Figure 1—figure supplement 1k*). Together, these results indicate that the process of human embryo polarization follows two steps: in the first step, F-actin becomes polarized and this happens concomitantly with embryo compaction, and in the second step, the PAR complex becomes polarized.

## Polarization of human embryos requires PLC signalling

We next wished to determine the mechanism of human embryo polarization. Our recent study in the mouse embryo showed that the onset of embryo polarization is driven by the PLC-Protein Kinase C (PKC) pathway, which enables the recruitment of the actin-myosin complex to the cell membrane (*Zhu et al., 2017*). To detect whether the PLC-PKC pathway is also responsible for the polarization of the human embryo, we first cultured human embryos from day 3 in the presence of the PLC inhibitor U73122 (*Bleasdale et al., 1990*), which we have previously used to block polarization in mouse embryos (*Zhu et al., 2017*). We tested three different concentrations (5 µM, 7.5 µM and 10 µM) of this inhibitor, and as controls, we used both untreated embryos and vehicle control embryos, treated with the same concentration of DMSO (35 mM DMSO as a control for 5 µM U73122, 53 mM DMSO as a control for 7.5 µM U73122 and 70 mM DMSO as a control for 10 µM U73122). Treatment with either 5 µM or 7.5 µM U73122 did not have an obvious effect on embryo cell numbers (*Figure 2—figure supplement 1b*). By contrast, high levels of PLC inhibitor (10 µM) affected embryo development (N = 66 control; N = 23 35 mM DMSO; N = 31 5 µM U73122; N = 20 53 mM DMSO; N = 20 7.5 µM U73122; N = 9 70 mM DMSO; and N = 10 10 µM U73122-treated embryos, *Figure 2—figure supplement 1c*).

Based on these results, all subsequent experiments were carried out using low U73122 concentrations (5 μM and 7.5 μM). We next analyzed the localization of F-actin and PARD6 on day 4 of development by immunofluorescence. Since there can be a batch effect in the number of polarized blastomeres in control untreated embryos from different experiments, which is likely due to patient-specific differences, control untreated embryos from multiple experiments were combined to carry out quantifications. We found that both 5 μM and 7.5 μM U73122 significantly reduced the apical enrichment of F-actin when compared to control embryos (N = 39 control embryos; N = 13 5 μM U73122-treated embryos; N = 23 7.5 μM U73122-treated embryos). However, when compared to the DMSO vehicle control the difference was not statistically significant (N = 7 35 mM DMSO-treated embryos; N = 15 53 mM DMSO-treated embryos) (*Figure 2—figure supplement 1e*). Similarly, the apical enrichment of PARD6 was not significantly different when U73122-treated embryos were compared to the DMSO vehicle control (*Figure 2—figure supplement 1f*).

Given the confounding effect of DMSO, we next used RNA interference (RNAi) (*Wianny and Zernicka-Goetz, 2000*) to down-regulate PLC expression at the cleavage stage and determine its effects on embryo polarization. Specifically, we depleted PLCE1 and PLCB1, as we found that their expression levels rank the highest among all functional PLC isoforms in human embryos between the two- to the eight-cell stage (*Figure 2—figure supplement 2aYan et al., 2013*). To test whether this experimental approach significantly decreased the levels of PLCE1 and PLCB1, we performed single-embryo qPCR of 3 -day-old embryos injected with siRNAs targeting these two genes at the zygote stage (*Figure 2—figure supplement 2b*, N = 18 embryos for control siRNA; N = 11 embryos for PLCB1+ PLCE1 siRNA embryos). These experiments revealed a 71% and 90% downregulation of PLCE1 and PLCB1 respectively (*Figure 2—figure supplement 2b*). Next, we injected control and PLCE1/PLCB1 siRNAs at the zygote stage and analyzed the localization of F-actin and PARD6 by immunofluorescence on day 4 (*Figure 2a–b*). The number of polarized cells and apical enrichment of PARD6 and in PLCE1/PLCB1 siRNA embryos was significantly reduced after depletion of PLCE1/ PLCB1 (N = 17 control siRNA embryos; N = 12 PLCB1+ PLCE1 siRNA embryos) (*Figure 2c–d*).

We also analyzed the effect of PLC inhibitor treatment and siRNA injection on the compaction of the human embryo. Both U73122 treatment and PLCE1+ PLCB1 siRNA led to a mild and non-statistically significant effect on compaction (N = 66 control; N = 23 35 mM DMSO; N = 31 5 μM U73122; N = 27 53 mM DMSO; N = 27 7.5 μM U73122; N = 14 control siRNA; and N = 13 PLCE1+ PLCB1 siRNA-injected embryos. *Figure 2—figure supplement 2c-d*). Together, these results indicate that PLC is a major regulator of cell polarization in the human embryo but not of compaction.

## Expression of TE markers is not initiated by cell polarization but reinforced by it

Finally, we wished to determine whether embryo polarization is critical for the initiation of cell differentiation into the TE, as is the case in the mouse embryo (*Jedrusik et al., 2008*). To address this question, we analyzed the spatiotemporal expression of the transcription factor GATA3, a key marker of TE specification that is already expressed at the compacting morula stage in human embryos (*Petropoulos et al., 2016*). We found no expression of GATA3 in human embryos after either 2 or 9 hr in culture, regardless of whether individual cells within the embryo were polarized or not (*Figure 3a–b*, N = 135 blastomeres; 17 embryos for 2 hr and N = 199 blastomeres; 20 embryos for 9 hr). In contrast, GATA3 expression became visible in 53.5 % of blastomeres after 24 hr of culture, on day 4 (*Figure 3a–b*, N = 519 blastomeres; 33 embryos). Remarkably, we found that GATA3 was expressed in both polarized and non-polarized blastomeres even at the early blastocyst stage, although the nuclear signal intensity of GATA3 was significantly higher in polarized cells (*Figure 3c* and *Figure 3—figure supplement 1a*). This is in contrast to the mouse, in which Gata3 expression becomes downregulated in the ICM at the early blastocyst stage (*Ralston et al., 2010*). These results suggest that in human embryos regulation of GATA3 expression might be initiated independently of embryo polarization, but reinforced by the acquisition of apicobasal polarity (*Figure 3a–c* and *Figure 3—figure supplement 1a*). To determine whether embryo polarization regulates other TE transcription factors, we also examined the expression of YAP1. We found that YAP1 was significantly enriched in polarized cells in comparison to unpolarized cells (N = 38 blastomeres; four embryos) (*Figure 3—figure supplement 1b-c*). These results suggest that cell polarization promotes the expression of TE-associated transcription at the morula stage but it is not essential for the initiation of their expression.

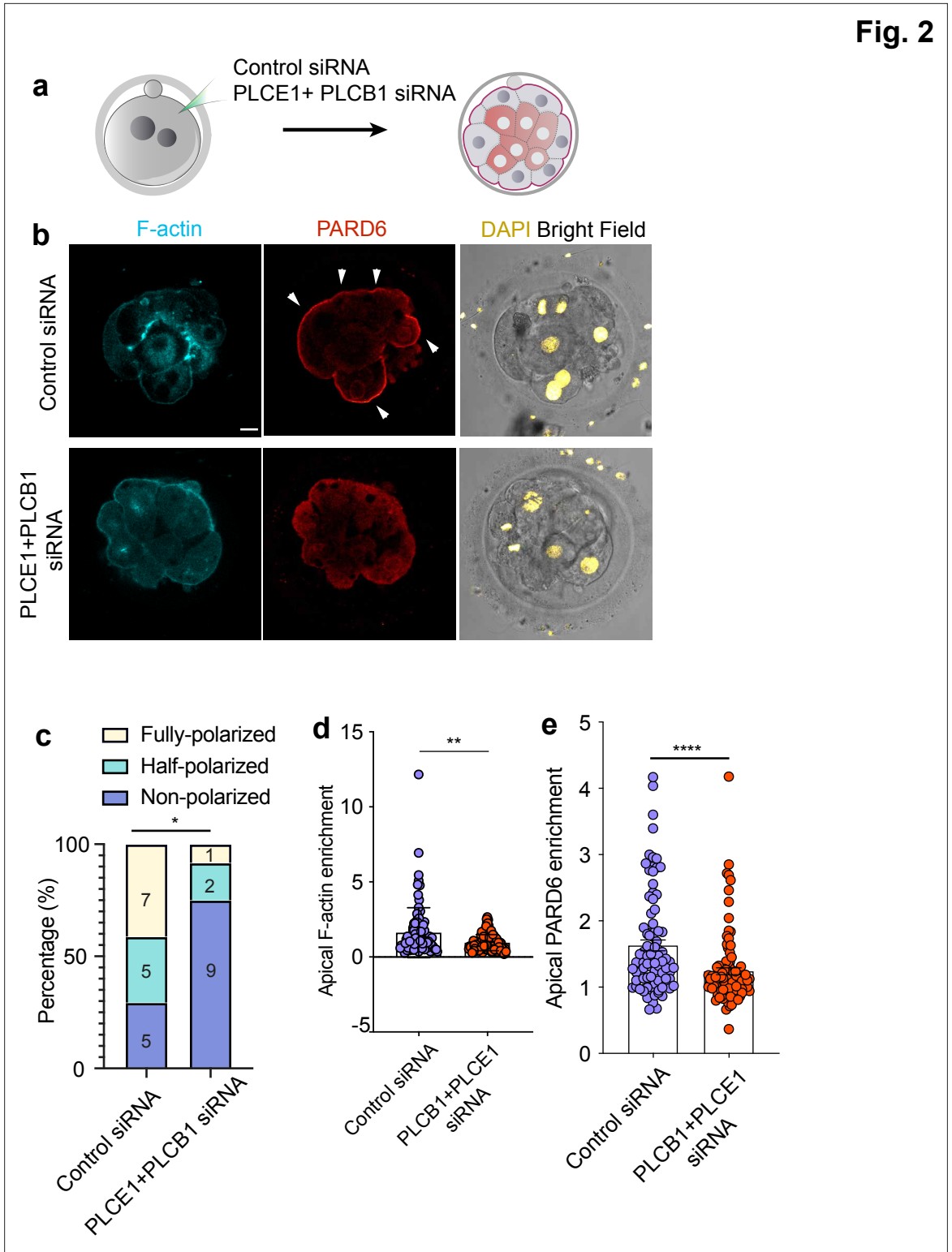

**Figure 2.** PLC activity regulates cell polarization in the human embryo. (**a**) Scheme of the PLCE1/PLCB1 siRNA injections. (**b**), Representative images of embryos injected with control siRNA or PLCE1/PLCB1 siRNA and cultured until embryonic day four to reveal the localization of F-actin, PARD6, and DAPI. (**c**) Quantification of the percentage of embryos showing PARD6 polarized cells in embryos from panel b. The number in each bar indicates the number of embryos analyzed. N = 17 embryos for control siRNA injected group; N = 12 for PLCB1+ PLCE1 siRNA injected group. * p < 0.05. Fisher's exact test. Three independent experiments. (**d**) Quantification of apical F-actin fluorescence intensity in embryos from panel b. N = 92 control siRNA cells and N = 100 PLCB1+ PLCB1 siRNA cells. **p < 0.0001, Mann-Whitney test. (**e**) Quantification of apical PARD6 fluorescence intensity in embryos

*Figure 2 continued on next page*

*Figure 2 continued*

from panel f. N = 89 control siRNA cells and N = 100 PLCB1+ PLCB1 siRNA cells. ****p < 0.0001, Mann-Whitney test. Scale bars, 15 μm.

The online version of this article includes the following figure supplement(s) for figure 2:

**Source data 1.** Source data for *Figure 2*.

**Figure supplement 1.** Morphokinetic analysis of the consequences of pharmacological inhibition of PLC.

**Figure supplement 2.** Morphokinetic analysis of PLC loss of function experiments.

To further confirm the role of cell polarization in the expression of TE transcription factors, we next wished to analyze GATA3 expression after PLC inhibitor treatment or depletion of PLCB1 and PLCE1. To this end, we applied U73122 at 5 μM and 7.5 μM, cultured embryos from day 3 to day 4, and analyzed GATA3 expression. We found that treatment with 7.5 μM U73122 led to a significant reduction of the nuclear GATA3 signal intensity when compared to control medium and 53 mM DMSO vehicle control (*Figure 3—figure supplement 1d-e*), N = 142 control blastomeres; N = 37 35 mM DMSO blastomeres; N = 84 53 mM DMSO blastomeres; N = 56 5 μM U73122 blastomeres; and N = 90 7.5 μM U73122 blastomeres. In agreement with the U73122 treatment results, depletion of PLCB1 and PLCE1 by siRNA also led to a significant reduction of GATA3 expression (*Figure 3d and e*). By classifying blastomeres according to their polarity status, we observed that the decrease in GATA3 expression was specific to unpolarized cells (*Figure 3f*), and hence the effect of PLCB1 and PLCE1 siRNA on GATA3 expression is through a reduction in the number of polarized cells. Together, these results suggest that the onset of GATA3 is controlled by a polarity-independent pathway, but the levels of nuclear GATA3 are reinforced by a polarity-dependent pathway.

The fact that expression of the TE lineage determinant, GATA3, is initiated independently of embryo polarization led us to examine when inner cells, precursors of the ICM, are first generated. In the mouse embryo, inner cells are produced after all blastomeres become polarized at the eight-cell stage and by subsequent cell divisions from the 8- to 16 -cell stage. Consequently, inner cells can only be observed in embryos having more than eight polarized cells. To determine whether the same is true in the human embryo, we first analyzed the number of inner cells and their correlation with cell polarity and overall cell numbers in fixed human embryos. Remarkably, we found that in 37 % of human embryos, inner cells were present with less than eight outer polarized cells (*Figure 4a–b*, N = 21 embryos). This suggests that the generation of inner cells in the human embryo might not require the preceding establishment of embryo polarization. In addition, we found that the size of inner cells was highly variable, ranging from 229.123 to 1896.4 μm$^2$ (*Figure 4c–d*). This partially overlapped with the size of eight-cell stage blastomeres, which ranges from 895.432 to 2627.03 μm$^2$ (*Figure 4d*). We found that in 34 % (18 out of 53) of the embryos we examined, inner cells were bigger than the smallest eight-cell stage blastomere. This result suggests cells might be internalized without undertaking asymmetric cell division at the 8- to 16 -cell stage transition. To assess this possibility, we performed time-lapse imaging of live fluorescently labeled human embryos to record cell positions during the process of embryo compaction. To be able to follow cells in live embryos, we labeled the cell membrane using a live-membrane dye (FM 4–64 FX). Our time-lapse recordings, allowed us to observe a case in which a blastomere became internalized during the process of embryo compaction (*Figure 4e*). Although we could only fluorescently label a small number of human embryos to record the spatial localization and size of cells, our results suggest that at least some cells are allocated to the ICM independently of the cell divisions of the eight-cell stage blastomeres in the human embryo.

## Discussion

The first cell fate decision is crucial for the successful development of the mammalian embryo as it segregates the embryonic and extra-embryonic lineages. In the mouse embryo, this first lineage segregation is controlled by polarity cues that become established at the eight-cell stage, but the mechanism and timing of embryo polarization during human development have remained unknown. Here, we show that the establishment of the apical domain in the human embryo follows two steps: in the first step, F-actin becomes polarized concomitantly to embryo compaction, while PAR complex polarization takes place in a second step. Moreover, RNAi treatment to deplete highly expressed PLC isoforms significantly reduced F-actin and PAR complex polarization. Thus, our results demonstrate

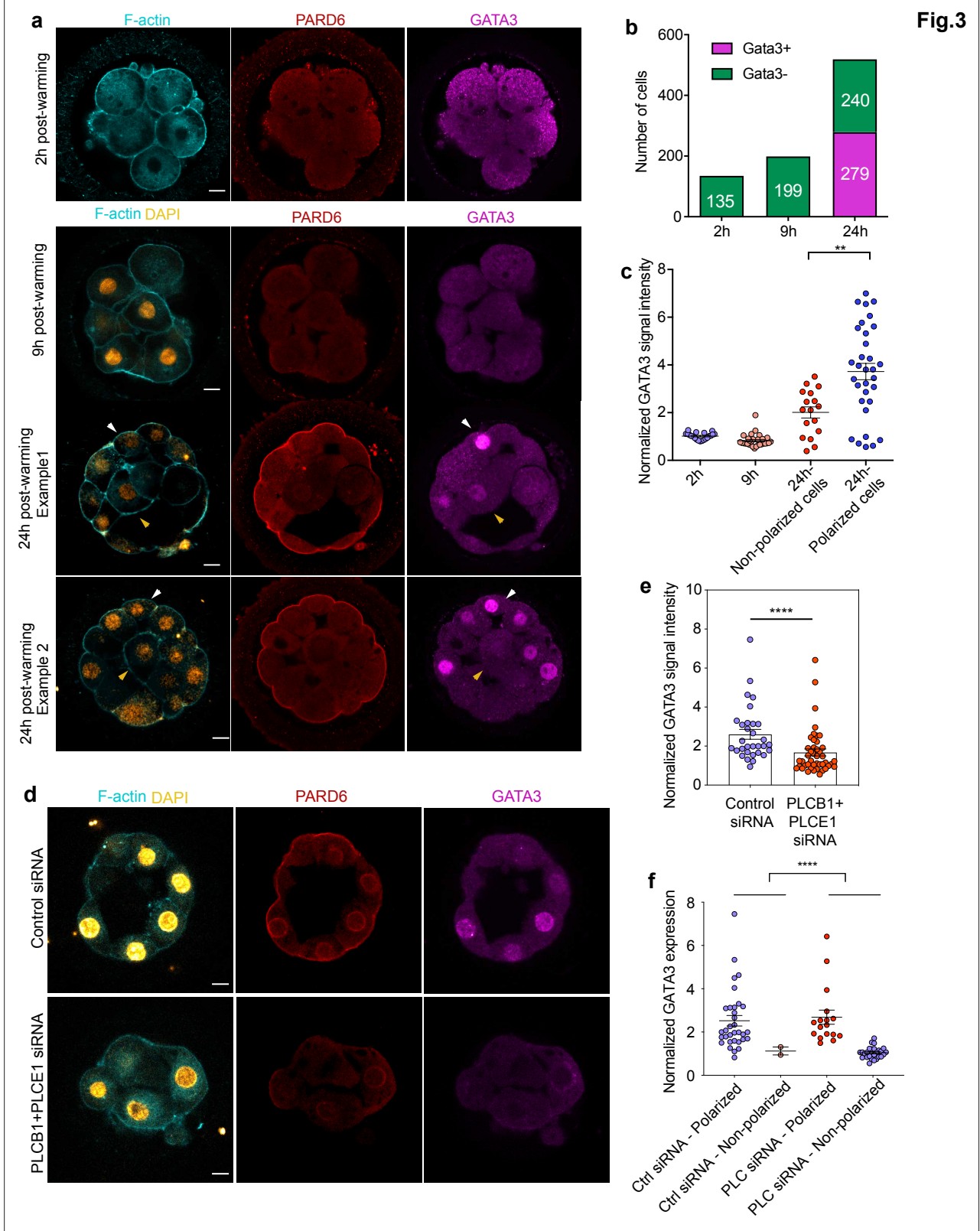

**Figure 3.** GATA3 expression is initiated independently of cell polarization. (**a**) Representative images of in vitro fertilized human embryos warmed at day 3 and cultured for 2, 9, or 24 hr (see scheme in *Figure 1a*) to reveal the localization of F-actin, PARD6, and GATA3. White arrowheads indicate outer cells; yellow arrowheads indicate inner cells. (**b**) Quantification of the number of GATA3-positive cells in embryos from panel a. Cells that have higher nuclear GATA3 signal (above a nucleus to cytoplasm ratio of 1.5) are categorized as GATA3+ cells. Numbers in each bar indicate the number of

*Figure 3 continued on next page*

*Figure 3 continued*

cells analyzed. ns, not significant, Fisher's exact test. N = 17 embryos for 2 hr N = 20 embryos for 9 hr and N = 33 embryos for 24 hr. (**c**) Quantification of GATA3 nuclear signal in embryos from panel a. Each dot represents one cell. Data are shown as mean ± S.E.M. The GATA3 nuclear signal has been calculated as the nucleus to cytoplasmic signal ratio in each cell. N = 21 cells (2 hr), N = 32 cells (9 hr), and N = 49 cells (24 hr). **p < 0.01; two-tailed unpaired Student's t-test. (**d**) Representative images of embryos injected with Control siRNA or PLCB1+ PLCE1 siRNA at the zygote stage, cultured until Day 4, and stained for F-actin, PARD6, and GATA3. (**e**), Quantification of GATA3 nuclear signal intensity (normalized against the cytoplasmic signal) in embryos from panel d. Each dot indicates one cell analyzed. N = 31 cells for Control siRNA group and N = 44 cells for PLCB1+ PLCE1 siRNA group. ****p < 0.0001, Mann-Whitney test. Five independent experiments (**a–c**) and two independent experiments (**d-f**). Scale bars, 15 μm. (**f**), Quantification of the level of nuclear GATA3 (normalized against cytoplasm signal) in embryos injected with Control siRNA or PLCB1+ PLCE1 siRNA. The data were classified according to the polarity status of the cells. Each dot indicates one cell. N = 31 cells for Control siRNA and N = 44 cells for PLCB1+ PLCE1 siRNA, ****p < 0.0001, Mann-Whitney test.

The online version of this article includes the following figure supplement(s) for figure 3:

**Source data 1.** Source data for *Figure 3* and *Figure 3—figure supplement 1*.

**Figure supplement 1.** Trophectoderm specification and inner cell generation in human embryos.

that PLC signaling is required for the establishment of cell polarity in the human embryo. Interestingly, the impairment of F-actin polarization upon PLC inhibition did not significantly compromise embryo compaction. Several mechanisms have been described to regulate compaction in the mouse embryo, including an actomyosin-mediated increase in cortical tension (*Maître et al., 2015*) and the formation of cadherin-enriched filopodia (*Fierro-González et al., 2013*). Our results, therefore, suggest that alternative pathways to F-actin-mediated cortical tension might be sufficient to drive compaction in the human embryo, which should be further tested in future studies.

In the mouse embryo, all blastomeres become compacted and polarized by the end of the eight-cell stage (*Johnson and Maro, 1984*). The results we present here indicate that the timing of compaction and polarization is more heterogeneous in human embryos as some human embryos complete compaction and polarization by the end of the eight-cell stage, while others complete these processes after the eight-cell stage. Although such developmental asynchronicity can be a species-specific feature of human embryo development, this variability could also reflect individual differences of the embryo donors, such as genetic factors or the age of the couples. In our studies, we have used supernumerary IVF human embryos from different clinics and countries, which could account for some variability across our experiments. To minimize experimental variations, we have used two pronuclei (2PN) embryos from the same IVF clinic for all descriptive experiments and pharmacological treatments. By necessity, for the PLC siRNA experiments, we used a mixture of 1PN, 2PN, and 3PN embryos which were randomly allocated to control and experimental groups.

The differences in the timing between completion of compaction and polarization may affect the way inner cells are generated. In mice, as embryos develop beyond the eight-cell stage, cells are allocated to inner positions through the cell divisions that take place after embryo compaction (*Anani et al., 2014*; *Samarage et al., 2015*). At this post-compaction stage, cell divisions generate polarized and non-polarized cells. Polarity then suppresses cellular contractility, leading to the internalization of non-polarized cells to the inside of the embryo (*Anani et al., 2014*; *Maître et al., 2016*; *Samarage et al., 2015*). Theoretically, the limited outer surface of the embryo restricts the number of outer cells (*Nissen et al., 2017*). In this context, if embryos compact with more than eight cells, some blastomeres may adopt an inner position due to the restricted surface space. As many human embryos compact with more than eight cells, this mechanism might apply to their development. In support of this possibility, our live imaging experiments revealed a case in which a blastomere became positioned to the inner compartment of the embryo during the process of compaction. Recent studies have demonstrated that already at the two-cell stage, blastomeres of the human embryo have a biased lineage allocation to either the ICM or the TE (*Bizzotto et al., 2021*; *Custers et al., 2021*), as previously shown in mouse embryos (*Gardner, 2001*; *Piotrowska et al., 2001*; *Piotrowska-Nitsche et al., 2005*). Future live imaging analyses of human embryos will be invaluable to dissect the relation between cleavage divisions, compaction, polarization, and lineage allocation during human embryogenesis.

In summary, our results uncover the timing and the mechanism leading to embryo polarization and its function in the first lineage segregation in the human embryo (*Figure 5*). During the revision of our manuscript, another article was published (*Gerri et al., 2020*) describing polarization events

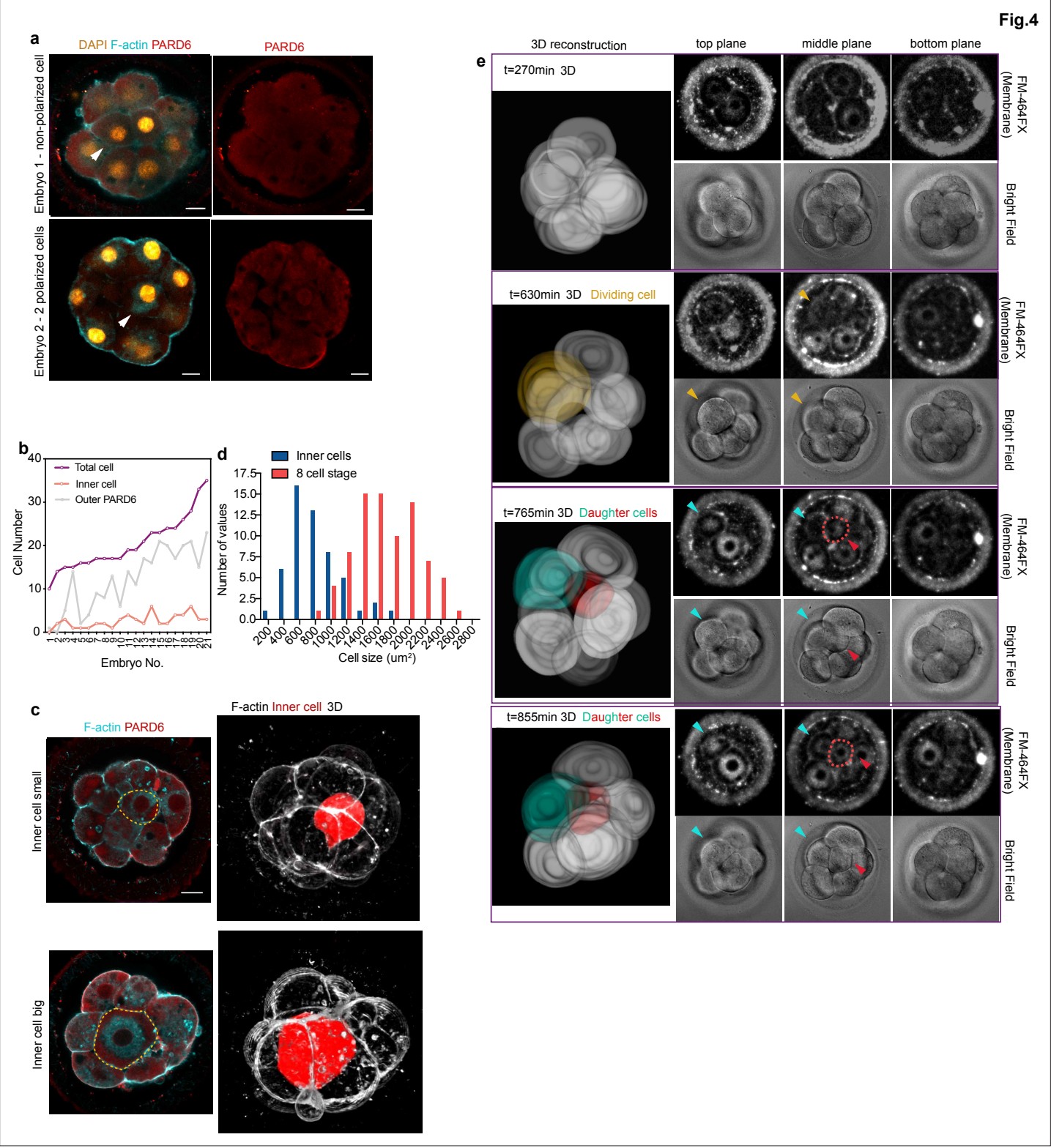

**Figure 4.** Generation of inner cells in the human embryo. (**a**) Representative images of embryos that have inner cells with a low number of outer polarized cells. White arrowheads indicate the presence of inner cells. (**b**) Line chart showing the relation between the number of inner cells, the number of PARD6 positive cells, and the total cell number in embryos that have inner cells. (**c**) Representative images of embryos showing inner cells of different sizes. Dotted lines indicate the outline of the inner cells. (**d**) Quantification of the distribution of the size of inner cells in comparison with the size of 8 cell stage blastomeres. N = 53 inner cells from 21 embryos; and N = 80, eight-cell stage blastomeres from 10 embryos. (**e**) Time-lapse imaging of cell

*Figure 4 continued on next page*

*Figure 4 continued*

position during human embryo compaction. After a cell division, one of the two daughter cells was positioned to the inside of the embryo. The dividing cell is labeled in yellow, and the two daughter cells are colored in green and red. Red dotted lines indicate the red cell shown in 3D reconstruction. The red cell becomes localized to the inside. Scale bars, 15 µm.

The online version of this article includes the following source data for figure 4:

**Source data 1.** Source data for *Figure 4*.

downstream of those we describe here in human embryogenesis. Consistent with our findings, Gerri and colleagues showed that cell polarity regulates TE differentiation in the human embryo as it does in the mouse. By characterizing earlier time points, our study demonstrates that the initiation of the expression of the TE transcription factor GATA3 is independent of cell polarization but that cell polarization reinforces the TE fate, which becomes established at the blastocyst stage. In addition, our results also identify activation of PLC signaling as the upstream mechanism that triggers polarization

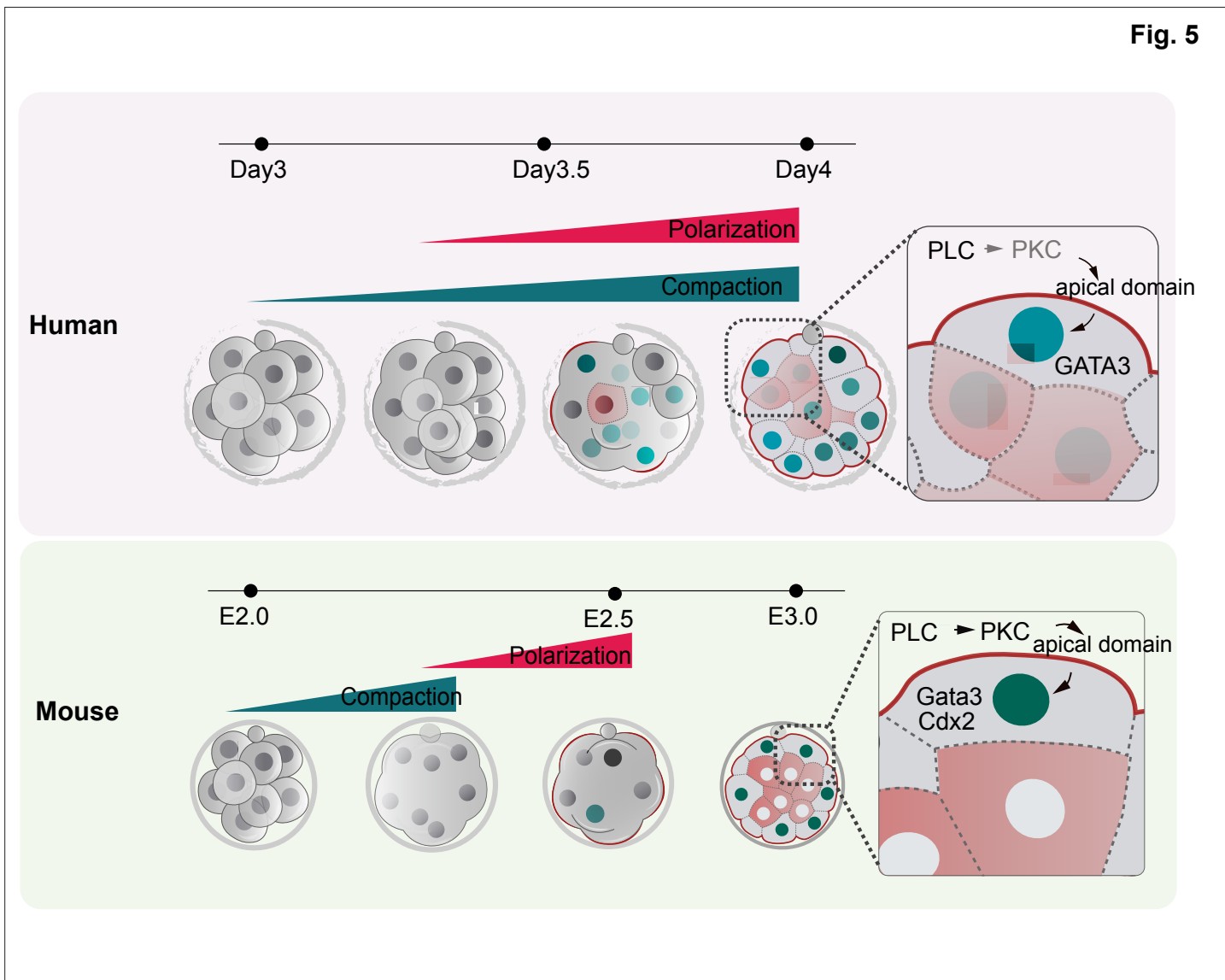

**Figure 5.** Schematic summarizing the main findings of this study. F-actin polarization precedes PAR complex polarization and it is triggered by PLC activation in mouse and human embryos. In human embryos, blastomeres initiate the expression of TE factors independently of the polarity machinery, but polarization reinforces a TE fate. In contrast to mouse embryos, in which inner cell generation takes place only once all blastomeres become polarized, inner cells can be observed in human embryos that are not fully polarized, indicating mechanistic differences in the first lineage decision between the two species.

in human embryos, providing mechanistic insight into human development at a stage when a crucial morphogenetic transition occurs.

# Materials and methods

**Key resources table**

| Reagent type (species) or resource | Designation | Source or reference | Identifiers | Additional information |
|---|---|---|---|---|
| Sequenced-based reagent | siRNA to PLCE1 and PLCB1 | Qiagen | Hs_PLCB1_4, SI00115521; Hs_PLCB1_6, SI02781184 Hs_PLCE1_1, SI00115521; negative control siRNA:1022076. | A 20 µM concentration of siRNA solution was used for injection. |
| Biological sample (Human) | Human embryos | Donated supernumerary embryos generated from in vitro fertilization experiments | | |
| Antibody | (Rabbit monoclonal), anti-PARD6 | Santa Cruz | sc-67393 | (1:200) |
| Antibody | (Goat polyclona)l, anti-GATA3 | R&D systems | AF2605 | (1:200) |
| Antibody | (Mouse monoclonal), anti-aPKC | Santa Cruz | sc-17781 | (1:50) |
| Antibody | (Mouse monoclonal), anti-YAP1 | Santa Cruz | sc-101199 | (1:200) |
| Recombinant DNA reagent | pRN3P- GAP-GFP | *Zhu et al., 2017* | | |
| Sequence-based reagent | GAPDH-F | This paper | qPCR primers | GATCATCAGCAATGCCTCCT |
| Sequence-based reagent | GAPDH-R | This paper | qPCR primers | TTCAGCTCAGGGATGACCTT |
| Sequence-based reagent | PLCB1-F | This paper | qPCR primers | GGAAGCGGCAAAAAGAAGCTC |
| Sequence-based reagent | PLCB1-R | This paper | qPCR primers | CGTCGTCGTCACTTTCCGT |
| Sequence-based reagent | PLCE1-F | This paper | qPCR primers | TGCAGCCTCTCATCCAGTT |
| Sequence-based reagent | PLCE1-R | This paper | qPCR primers | CCCTGCGGTAAATAGTCTGC |
| Commercial assay or kit | SMART-Seq v4 Ultra Low Input RNA Kit | Takara | Cat. No. 634,888 | |
| Commercial assay or kit | Agencourt AMPure XP Kit | Beckman Coulter | A63880 | |
| Chemical compound, drug | U73122 | Caymanchem | No. 70,740 | |
| Chemical compound, drug | DMSO | Sigma-Alrich | D2650−5 × 10 ML | |
| Software, algorithm | Prism 8 | Graphpad | | |

## Ethical approval

This study was performed in five different avenues: Clinical Embryology Laboratory at IVIRMA Valencia (Spain), University of Cambridge (United Kingdom), National Research Center for Assisted Reproductive Technology and Reproductive Genetics (China), California Institute of Technology (United States), and Foundation for Embryonic Competence New Jersey (United States). All the work complies with The International Society for Stem Cell Research (ISSCR) guidelines (*Lovell-Badge et al., 2021*).

### Experiments performed in Valencia (Spain)

The work in the Clinical Embryology Laboratory at IVIRMA Valencia was approved by the National Commission of Human Reproduction (CNRHA), the General direction of research, innovation, technology, and quality, and the ethics committee of Clinical Research IVI Valencia, which complies with Spanish law on assisted reproductive technologies (14/2006). A total of 260 supernumerary and cryopreserved donated day three human embryos from 95 IVF patients were used. Embryos were not created for research. Couples donating their embryos were informed of the objectives of the project, the nature of the experiments, and the conditions of the research. No financial inducements were offered for donation. The average age of women was 28.12 ± 4. The fixed embryos were analyzed at the University of Cambridge. This work was evaluated by the Human Biology Research Ethics Committee of the University of Cambridge (reference HBREC.2017.27).

### Experiments performed in Cambridge (UK)

The work performed at the University of Cambridge was in accordance with the Human Fertility and Embryology Authority (HFEA) regulations (license reference R0193). Ethical approval was obtained from the 'Human Biology Research Ethics Committee' of the University of Cambridge (reference HBREC.2017.21). Informed consent was obtained from all patients from CARE Fertility Group and Herts & Essex fertility clinics, who donated supernumerary and cryopreserved embryos after completing their IVF treatment. Embryos were not created for research. Before giving consent, patients were informed about the specific objectives of the project, and the conditions that apply within the license, they were offered counseling and did not receive any financial inducements. In this study, we used 15 donated embryos at the two pronuclei stage (day 1 d.p.f.), which were warmed and cultured in the Cambridge Laboratory according to the above regulations.

### Experiments performed in Shandong (China)

The work performed in the National Research Center for Assisted Reproductive Technology and Reproductive Genetics was conducted under the regulations of the Human Biomedical Research Ethics Guidelines (regulated by the National Health Commission of the People's Republic of China on 1 December 2016) and the Human Embryonic Stem Cell Research Ethics Guidelines (regulated by China National Center for Biotechnology Development on 24 December 2003). These regulations and guidelines allow 3PN human gametes, and/or human embryos created or genetically manipulated in vitro and those cultured for no more than 14 days, to be used for scientific researches. The aim and protocols involved in this study were reviewed and approved by the Institutional Review Board (IRB) of Reproductive Medicine, Shandong University. The protocols include the use of 3PN embryos and siRNA injections. All embryos were donated after informed consent was obtained. The embryos were supernumerary and fresh 3PN zygotes generated from in vitro fertilization, with the donor female age ranging from 22 to 40 years old. These embryos were used for microinjection experiments. Embryos were not created for research. Patients were informed about the objectives, experimental approach, and potential outcomes of the research before the donation. They were offered counseling to aid them in their decision.

### Experiments performed in California (US)

The work at the California Institute of Technology was approved by the California Institute of Technology Committee for the Protection of Human Subjects (IRB number 19–0948). Human embryos at the zygote stage were obtained from the University of Southern California (USC) through the pre-existing USC IRB-approved Biospecimen Repository for Reproductive Research (HS-15–00859) after appropriate approval was obtained unanimously from the Biorepository Ethics Committee. At USC Fertility, supernumerary cryopreserved embryos were donated after completion of IVF; embryos were

not created for research purposes. Patients were informed of the general conditions of the donation, objectives, and methodology of human embryo research. They were offered counseling and alternatives, including discarding embryos and continued cryopreservation. Patients were informed that they would not benefit directly from the donation of embryos to research. A total of 41 donated human embryos at the zygote two pronuclei stage (day 1 post-fertilization) from four IVF patients were used. The average age of women was 44. The embryos were warmed at USC Fertility per the usual IVF procedure and then transferred to the California Institute of Technology for the remainder of the protocol.

### Experiments performed in New Jersey (United States)

The research was approved by the Western IRB (Clinical IRB 20031397). Embryos used in this study were supernumerary fresh embryos for the IVF treatment of IVI-RMA patients. Informed consent was obtained from all the couples that donated their embryos. They were informed of the general objective, methodology, and potential outcomes of the research. Patients did not receive any financial compensation for the donation. Embryos were not created for research.

## Embryo selection, embryo warming, and culture conditions

### Experiments performed in Valencia (Spain)

The majority of the day three embryos used in this study were generated by intracytoplasmic sperm injection (ICSI) (76.2% = 198/260) using donor eggs (72.7% = 189/260). The mean age of female patients providing oocytes was 28.12 ± 4 years (N = 260), which is one of the strongest predictors of embryonic competence and oocyte quality (*Cimadomo et al., 2018*). The mean age of male patients providing sperm was 40.8 ± 6.98 years (N = 260). The mean cell number at the time of embryo warming was 7.98 ± 1.26 (N = 260), which is consistent with the eight cells expected on the day-3 stage of development of a good quality human embryo (*Reproductive and Embryology, 2011*). Most common female indications for Assisted Reproductive Technology included age (55% = 143/260), poor ovarian response (8.1% = 21/260), and tubal factor (7.7% = 20/260).

Vitrification and warming procedures were performed using the Kitazato method, as described elsewhere (*Cobo et al., 2010*; *Kuwayama, 2007*). Briefly, embryos were equilibrated at room temperature in 7.5 % (vol/vol) ethylene glycol +7.5 % dimethylsulfoxide (DMSO). After volume re-expansion, embryos were transferred to the vitrification solution consisting of 15 % ethylene glycol +15 % DMSO + 0.5 M sucrose. After 1 min in this solution, embryos were placed on the Cryotop device using a minimum volume and were directly submerged in liquid nitrogen. For warming, the Cryotop was removed from liquid nitrogen and placed in 1.0 M sucrose in tissue culture media M 199 + 20 % synthetic serum substitute (SSS) at 37 °C. After 1 min, embryos were transferred to a solution containing 0.5 M sucrose at room temperature for 3 min. After two washes of 5 and 1 min, each in TCM199, embryos were cultured in a Geri Dish at 5.5 % $CO_2$, 5 % $O_2$, and 37 °C, in a humidified environment in a time-lapse incubator Geri (Genea Biomedx, Australia). Embryos were cultured for 2, 9, or 24 hr in pre-equilibrated Sydney IVF Blastocyst Medium (Cook, USA) or in the same medium supplemented with either U73122 - a PLC inhibitor (Caymanchem, USA, in 35 mM or 53 mM DMSO) or DMSO (35 mM or 53 mM, Sigma, USA) as vehicle control, without mineral oil. All culture media were pre-equilibrated to the incubator's conditions overnight (5.5 % $CO_2$, 5 % $O_2$, and 37 °C, humidified environment).

### Experiments performed in Cambridge (UK)

Cryopreserved day one embryos were warmed using Origio thaw kit (REF10984010A) following the manufacturer's instructions. Briefly, the day before warming, the Global Total human embryo culture medium (HGGT-030, LifeGlobal group) was incubated at 37 °C + 5% $CO_2$. Upon warming, the straw containing the embryo was immersed in prewarmed (37 °C) water for 1 min. The embryo was then transferred into vial 1 (5min), vial 2 (5 min), vial 3 (10 min), and finally in vial 4 for the slow warming procedure. All these incubation steps were done using 12 well plates (ThermoFisher, 150628) and 1 ml per solution. Warmed embryos were finally incubated in drops of pre-equilibrated Global Total human embryo culture medium under mineral oil (9305, Irvine Scientific). Culture conditions are the following: 37 °C, 21 % $O_2$, and 5 % $CO_2$. Embryos were incubated for a total of 48 hr in the Global reaching 6–8 cell stage (day 3). On day 3, the embryos were either transferred into a 96-well plate

with each well containing 200 µl solution: control group (Global medium only), treated group (Global medium+ PLC inhibitor at the concentration of 7.5 µM), or transferred to medium containing 35 ng/ml FM 4–64 FX for live-imaging. Embryos were fixed with 4 % PFA after 24 hr of culture.

### Experiments performed in California (US):

Embryos were warmed using Embryo Thaw Media Kit following the manufacturer's instructions (Fujifilm Irvine Scientific, Cat. No. 90124). The day before warming, Continuous Single Culture-NX Complete medium (90168, Fujifilm Irvine Scientific) was equilibrated overnight at 37 °C, 5 % $CO_2$. On the day of warming (day 1), the straw that contains the embryo was defrosted at room temperature for 30 s and immersed in prewarmed (37 °C) water for 1 min until the ice melted. The embryo was then transferred into T-1 (5 min), T-2 (5 min), T-3 (10 min) solutions for slow warming, and finally into Multipurpose Handling Medium (90163, Fujifilm Irvine Scientific) for recovery. All these incubation steps were done using four well plates (Nunc) and 1 ml per solution. Warmed embryos were finally incubated in drops of pre-equilibrated Continuous Single Culture-NX Complete medium under mineral oil (9305, Irvine Scientific). Culture conditions are the following: 37 °C, 21 % $O_2$, and 5 % $CO_2$. Embryos were incubated for a total of 48 hr until reaching the morula stage (day 4). Embryos were fixed with 4 % PFA at day 4 for immunofluorescence analysis.

### Experimental Setting

A total of 70 day-three embryos were cultured for 2 (N = 17 embryos), 9 (N = 20 embryos), and 24 hr (N = 33 embryos) after warming. Following fixation in 4 % PFA, the compaction and polarization timings of the human embryos were assessed by immunofluorescence analysis and time-lapse imaging evaluation.

To test the role of PLC, eight experiments were performed with a total of 207 day-three human embryos, which were warmed and randomly allocated into one of the following experimental groups: culture medium control (N = 66), 5 µM U73122 (N = 31), and its vehicle control (35 mM DMSO, N = 23), 7.5 µM U73122 (N = 34), and its vehicle control (53 mM DMSO, N = 34), and 10 µM U73122 (N = 10), and its vehicle control (70 mM DMSO, N = 9). All embryos were cultured for 24 hr and fixed in 4 % paraformaldehyde (PFA) to analyze cell polarization and TE specification.

Embryo development was recorded in a Geri incubator (Genea Biomedx, Australia), which was programmed to acquire images of each embryo every 15 min through 11 different focal planes. The time-lapse videos of embryo development were analyzed using the manual annotation software Geri Assess 1.0 (Genea Biomedx, Australia). The morphokinetic analysis was performed from the moment of embryo warming (Day 3) until embryo fixation at 2, 9, or 24 hr after warming.

### Human embryo microinjection

The following siRNAs were used for microinjection: human PLCB1 siRNA #1 (CACACTACCAAG TATAATGAA) (Qiagen, Hs_PLCB1_4, SI00115521); PLCB1 siRNA #2 (CAGAGATGATCGGTCATATA) (Qiagen, Hs_PLCB1_6, SI02781184); PLCE1 siRNA#1(CAGGGTCTTGCCAGTCGACTA) (Qiagen, Hs_PLCE1_1, SI00115521); negative control siRNA (UUCUCCGAACGUGUCACGUdTdT) (Qiagen, 1022076). A 20 µM concentration of siRNA solution was used for injection.

### Experiments performed in China

For the siRNA microinjection experiment performed at the National Research Center for Assisted Reproductive Technology and Reproductive Genetics, IVF 3PN zygotes were injected and cultured in G1.5 medium (Vitrolife, cat. no.10128).

### Experiments performed in California (United States)

for the siRNA microinjection experiment performed at the California Institute of Technology, the warmed embryos at the 2PN stage were placed in MHM, and microinjections were performed in MHM. Embryos were cultured in drops of pre-equilibrated medium overlaid with mineral oil as described above.

### Experiments performed in New Jersey (United States)

Microinjection was performed on fresh IVF zygotes (1PN or 3PN). Zygotes were suspended in drops of G-MOPS PLUS (Vitrolife) covered by OVOIL (Vitrolife) on a Falcon culture dish. Both holding (Humagen Micropipets, Origio) and injection pipettes were held in micromanipulators (Nikon). The injection pipette was made from capillaries (Harvard Apparatus) pulled on an automatic micropipette puller (Sutter). The siRNA was loaded into the back of the capillary and the microinjection was performed with an Eppendorf Femtojet microinjector (injection pressure = 300 hPa; injection time = 0.10 sec; compensation pressure = 40 hPa). After injection, zygotes were transferred to G1 PLUS (Vitrolife) under OVOIL and cultured at 37 °C, 6 % $CO_2$, and 5 % $O_2$ until the third day of development.

## Embryo fixation and immunofluorescence

After culture, embryos were washed in PBS (D8537, Sigma) and fixed in a freshly prepared PBS solution containing 4 % PFA (15710, EMS) for 20 min. For experiments done in IVIRMA Valencia, fixed embryos were washed twice in a PBS solution containing 0.1 % Tween-20 (P9416, Sigma) and immediately placed into a 0.5 ml PCR tube within oil-PBS-oil interphase. Tubes were stored at 4 °C were shipped to the University of Cambridge for immunofluorescence. For experiments done in other avenues, the embryos were directly processed for the subsequent steps.

Embryos were permeabilized in PBS containing 0.3 % Triton X-100 and 0.1 M glycine for 20 min at room temperature. They were incubated in blocking buffer (PBS containing 10 % BSA and 0.1 % Tween) for 1 hr. Primary antibodies were incubated overnight at 4 °C in blocking buffer and secondary antibodies were incubated at room temperature for 2 hr in blocking buffer. The following antibodies were used:

Primary antibodies:

Rabbit anti-PARD6 (1:200, Santa Cruz, sc-166405);
Goat anti-GATA3 (1:200, Thermo Fisher, MA1-028);
Mouse anti-aPKC (1:50, Santa Cruz, sc-17781);
Mouse anti-YAP1 (1:200, Santa Cruz, sc-101199).
Secondary antibodies:
Alexa Fluor 568 Donkey anti-Goat;
Alexa Fluor 568 Donkey anti-Mouse;
Alexa Fluor 647 Donkey anti-Rabbit.
Alexa Fluor 647 Donkey anti-mouse.

In addition, embryos were stained with Alexa Fluor 488 Phalloidin (A12379, ThermoFisher Scientific) to reveal the F-actin cytoskeleton and with DAPI (D3571, ThermoFisher Scientific) to reveal nuclei.

Images were taken on an SP5 confocal microscope (Leica Microsystems) using a 40 x oil objective N.A = 1.2. Laser power (less than 10%) and gain were kept consistent across different samples from the same experiment.

The images provided for siRNA injection experiments were taken on either an Andor Dragonfly spinning disc confocal microscope (63 x oil objective N.A = 1.4), or SP8 confocal microscopy (Leica Microsystems) using a 40 x oil objective N.A = 1.10.

## RNA extraction and qRT-PCR validation

Each embryo was collected in 1 µl of lysis buffer. RNA was prepared using the SMART-Seq v4 Ultra Low Input RNA Kit, according to the user manual, and the cDNA was amplified by Long Distance (LD) PCR (20 cycles). Complementary DNA (cDNA) was purified using the Agencourt AMPure XP Kit (Beckman Coulter). The concentration of the cDNA was measured by Qubit using a dsDNA High sensitivity kit. Real-time quantitative polymerase chain reaction (RT-qPCR) was performed to confirm the silencing of the PLCE1 and PLCB1 genes in the siRNA injected embryos. The PCR reaction was prepared using 5 µL PowerUpTM SYBRTM Green Master Mix (2 x), 1 µL forward and reverse primers, 1 µL of DNA template, and 3 µL of nuclease-free water.  Five ng of cDNA was used per reaction. The samples were run in duplicates. ViiA7 RT-qPCR machine by Applied Biosystems was used for the PCRs. The components were mixed thoroughly and briefly centrifuged. PCR cycling conditions were 95 °C for 2 min followed by 40 cycles of 15 s at 95 °C, 58 °C for 60 s. GAPDH was used as a housekeeping gene. The $\Delta\Delta$Ct method was used for the calculation of the difference in the expression of genes. A melt curve analysis was performed to confirm the specificity of products.

The following primers have been used for RT-qPCR: GAPDH: forward: GATCATCAGCAAT-GCCTCCT; reverse: TTCAGCTCAGGGATGACCTT; PLCB1: forward: GGAAGCGGCAAAAAGA AGCTC; reverse: CGTCGTCGTCACTTTCCGT; PLCE1: forward: TGCAGCCTCTCATCCAGTT; reverse: CCCTGCGGTAAATAGTCTGC.

## Data analysis

Time lapse imaging: Since the exact times of cryopreservation were not available in most cases, the following variables were assessed from the time of warming (t0) by means of time-lapse imaging: number of intact and degenerated cells, defined as the number of apparent viable and degenerated cells, respectively; number of total cells, defined as the sum of intact and degenerated cells; degree of fragmentation, defined as the estimated percentage of the embryonic volume occupied by cellular fragments; number of degenerating cells; defined as the number of cells degenerating during embryo culture; cell cycle length, defined as the time in hours between two consecutive cytokinesis in a given blastomere; time to the start of compaction, defined as the first time point after warming at which two adjacent blastomeres displayed an increased inter-blastomere angle; time to the end of compaction, defined as the first time point after warming at which all blastomeres displayed an increased inter-blastomere angle; duration of compaction, defined as the time interval between the start and end of compaction; number of cells at the start and end of compaction; and number of cells after embryo culture. Cell numbers shown in Figure 1 - figure supplement 1 have been obtained using time-lapse images. To calculate cell numbers after compaction and after embryo culture, the number of cell divisions throughout the compaction stage and the whole embryo culture were added to the number of cells before compaction and to the number of cells after warming, respectively. Hence, since some cell divisions may be untraceable when occurring inside the compacting embryonic mass, real cell numbers may be higher than reported.

All immunofluorescence images were analyzed using Fiji (*Schneider et al., 2012*), Rasband, & Eliceiri, 2012 .

### Apical enrichment analysis

F-actin and PARD6 polarization were measured in a single focal plane, by taking the middle plane of the embryo. A freehand line of the width of 0.5 μm was drawn along the cell-contact free surface (apical domain), or cell-contact (basal) area of the cell and signal intensity was obtained via the Region of Interest (ROI) function of Fiji. The apical/basal signal intensity ratio is calculated as I(apical)/I(basal). A cell is defined as polarized when the ratio between the apical membrane and the cytoplasm signal intensity exceeds 1.5.

### Expression analysis

The nucleus of each cell was masked using the Region of Interest (ROI) tool of Fiji. The average signal intensity of the ROI is calculated and a cell is defined as GATA3 positive when the nucleus to cytoplasm signal intensity exceeds 1.5.

### Cell position analysis

Embryos were imaged across their entire Z volume. By looking at F-actin localization, the area of the cell of interest was examined on each plane. Cells that had an area exposed to the outside were defined as outer cells, whereas cells that did not have an area exposed to the outside were defined as inner cells.

### Cell size measurements

We analyzed the size of blastomeres in 8-cell stage human embryos (reference) and in human embryos harboring inner cells. Cell size was measured in a representative single plane using the F-actin channel and manually drawing the contour of the cells.

### Inter-blastomere angle

To measure the angle between two adjacent blastomeres we used the angle tool in Fiji and applied it to interfaces between blastomeres that were visible in a given Z plane. 120° was chosen as a cut-off. The initiation of compaction was established as the time point at which one of the inter-blastomere

angles in an embryo was above 120°. The end of completion of compaction was established as the time point at which all inter-blastomere angles in an embryo were above 120°.

### Reconstruction and segmentation

For the 3D reconstruction images provided in figure 4, the membrane region of each cell at each plane was determined by FM-464FX signal, and the region of a cell across all planes was segmented manually using Fiji. The segmented masks were then transferred to 3D slicer software for the generation of 3D reconstructed images. Statistics and reproducibility.

Statistical analyses were done using GraphPad Prism. Investigators were not blind to group allocation. The sample size was not pre-defined and embryos were randomly allocated into different experimental groups. Qualitative data was analyzed using a Chi-square test. Quantitative data that presented a normal distribution were analyzed using a two-tailed Student's t-test or an ANOVA test. Quantitative data that did not present a normal distribution were analyzed using a Mann Whitney U test or a Kruskal Wallis test.

## Acknowledgements

We are thankful to David Glover for comments on the manuscript, Marta Perez Sanchez, Antonia Weberling, Bailey Weatherbee, and Ali Ahmady for the help with human embryo culture, and to USC Fertility and HCLD for their support. M.Z. is funded by Leverhulme Trust. M.N.S. is funded by the European Molecular Biology Organization (EMBO) and the Medical Research Council (MRC, MC_UP_1201/24). Work in the laboratory of M.Z-G. on human embryos is funded by Wellcome Trust (207415/Z/17/Z), Open Philanthropy Grant, Curci, and Weston Havens Foundations. Work in the laboratory of Zi-Jiang Chen is funded by The National Key Research and Development Program of China (2018YFC1004000) and Shandong Provincial Key Research and Development Program (2018YFJH0504).

## Additional information

### Funding

| Funder | Grant reference number | Author |
|---|---|---|
| Wellcome Trust | 207415/Z/17/Z | Magdalena Zernicka-Goetz |
| Open Philanthropy Project | | Magdalena Zernicka Goetz |
| Curci and Weston Heavens Foundations | | Magdalena Zernicka-Goetz |
| Leverhulme Trust | RPG-2018-085 | Meng Zhu |
| European molecular biology organisation | | Marta Shahbazi |
| Medical Research Council | MC_UP_1201/24 | Marta Shahbazi |
| National Key Research and Development Program of China | 2018YFC1004000 | Zijiang Chen |
| Shandong Provincial Key Research and Development Program | 2018YFJH0504 | Zijiang Chen |

The funders had no role in study design, data collection and interpretation, or the decision to submit the work for publication.

### Author contributions

Meng Zhu, Marta Shahbazi, Formal analysis, Investigation, Methodology, Validation, Visualization, Writing – original draft, Writing – review and editing; Angel Martin, Formal analysis, Investigation, Methodology, Validation, Visualization; Chuanxin Zhang, Berna Sozen, Mate Borsos, Marga Esbert, Shiny Titus, Investigation, Validation; Rachel S Mandelbaum, Investigation; Richard J Paulson,

Resources, Supervision; Matteo A Mole, Investigation, Visualization; Richard T Scott, Alison Campbell, Simon Fishel, Keliang Wu, Resources; Viviana Gradinaru, Funding acquisition, Supervision; Han Zhao, Zi-Jiang Chen, Funding acquisition, Resources, Supervision; Emre Seli, Conceptualization, Funding acquisition, Resources, Supervision; Maria J de los Santos, Conceptualization, Formal analysis, Funding acquisition, Investigation, Methodology, Resources, Supervision; Magdalena Zernicka Goetz, Conceptualization, Funding acquisition, Methodology, Resources, Supervision, Writing – original draft, Writing – review and editing

**Author ORCIDs**
Meng Zhu (iD) http://orcid.org/0000-0001-6157-8840
Marta Shahbazi (iD) http://orcid.org/0000-0002-1599-5747
Viviana Gradinaru (iD) http://orcid.org/0000-0001-5868-348X
Zi-Jiang Chen (iD) http://orcid.org/0000-0001-6637-6631
Emre Seli (iD) http://orcid.org/0000-0001-7464-8203
Magdalena Zernicka Goetz (iD) http://orcid.org/0000-0002-7004-2471

**Ethics**
Human subjects: Ethical approval This study was performed in five different avenues: Clinical Embryology Laboratory at IVIRMA Valencia (Spain), University of Cambridge (United Kingdom), National Research Center for Assisted Reproductive Technology and Reproductive Genetics (China), California Institute of Technology (United States), and Foundation for Embryonic Competence New Jersey (United States). All the work complies with The International Society for Stem Cell Research (ISSCR) guidelines. Experiments performed in Spain: The work in the Clinical Embryology Laboratory at IVIRMA Valencia was approved by the National Commission of Human Reproduction (CNRHA), the General direction of research, innovation, technology and quality and the ethics committee of Clinical Research IVI Valencia, which complies with Spanish law on assisted reproductive technologies (14/2006). A total of 260 excess and cryopreserved donated day 3 human embryos from 95 IVF patients were used. Embryos were not created for research. Couples donating their embryos were informed of the objectives of the project, the nature of the experiments, and the conditions of the research. No financial inducements were offered for donation. The average age of women was 28.12 ± 4. The fixed embryos were analysed at the University of Cambridge. This work was evaluated by the Human Biology Research Ethics Committee of the University of Cambridge (reference HBREC.2017.27). Experiments performed in the United Kingdom: The work performed at the University of Cambridge was in accordance with the Human Fertility and Embryology Authority (HFEA) regulations (license reference R0193). Ethical approval was obtained from the "Human Biology Research Ethics Committee" of the University of Cambridge (reference HBREC.2017.21). Informed consent was obtained from all patients from CARE Fertility Group and Herts & Essex fertility clinics, who donated excess and cryopreserved embryos used in this study after completing their IVF treatment. Embryos were not created for research. Prior to giving consent, patients were informed about the specific objectives of the project, and the conditions that apply within the license, offered counselling and did not receive any financial inducements. In this study, we used 15 donated embryos at the 2 pronuclei stage (day 1 d.p.f.), which were warmed and cultured in the Cambridge Laboratory according to the above regulations. Experiments performed in China: The work performed in the National Research Center for Assisted Reproductive Technology and Reproductive Genetics was conducted under the regulations of the Human Biomedical Research Ethics Guidelines (regulated by National Health Commission of the People's Republic of China on 1 December 2016) and the Human Embryonic Stem Cell Research Ethics Guidelines (regulated by China National Center for Biotechnology Development on 24 December 2003). These regulations and guidelines allow excess 3PN human gametes, and/or human embryos created or genetically manipulated in vitro and those cultured for no more than 14 days, to be used for scientific researches. The aim and protocols involved in this study were reviewed and approved by the Institutional Review Board (IRB) of Reproductive Medicine, Shandong University. The protocols include the use of 3PN embryos and the siRNA injections. All embryos were donated after informed consent was obtained. The embryos were excess and fresh 3PN zygotes generated from in vitro fertilisation, with the donor females age ranging from 22 to 40 years old. These embryos were used for microinjection experiments. Embryos were not created for research. Patients were informed about the objectives, experimental approach and potential outcomes of the research prior to the donation. They were offered counselling to aid them in their decision. Experiments performed

in California (United States): The work at California Institute of Technology was approved by the California Institute of Technology Committee for the Protection of Human Subjects (IRB number 19-0948). Human embryos at the zygote stage were obtained from the University of Southern California (USC) through the pre-existing USC IRB-approved Biospecimen Repository for Reproductive Research (HS-15-00859) after appropriate approval was obtained unanimously from the Biorepository Ethics Committee. At USC Fertility, excess cryopreserved embryos were donated after completion of IVF; embryos were not created for research purposes. Patients were informed of the general conditions of the donation, objectives, and methodology of human embryo research. They were offered counselling and alternatives, including discarding embryos and continued cryopreservation. Patients were informed that they would not benefit directly from donation of embryos to research. A total of 41 donated human embryos at the zygote 2 pronuclei stage (day 1 postfertilisation.) from 4 IVF patients were used. The average age of women was 44. The embryos were warmed at USC Fertility per usual IVF procedure and then transferred to the California Institute of Technology for the remainder of the protocol. Experiments performed in New Jersey (United States): the research was approved by the Western IRB (Clinical IRB 20031397). Embryos used in this study were excess fresh embryos to the IVF treatment of IVI-RMA patients. Informed consent was obtained from all the couples that donated their embryos. They were informed of the general objective, methodology and potential outcomes of the research. Patients did not receive any financial compensation for the donation. Embryos were not created for research.

## Decision letter and Author response
Decision letter https://doi.org/10.7554/eLife.65068.sa1
Author response https://doi.org/10.7554/eLife.65068.sa2

## Additional files

### Supplementary files
• Transparent reporting form

### Data availability
Source data files used for generating each figure has been uploaded as supplementary material.

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
