## [Decision Letter]

**Acceptance summary:**

This paper painstaking explores the basis for the first lineage decisions in human embryogenesis, the segregation of trophectoderm from inner cell mass. The mechanisms explored here provide insight into the process of compaction and polarization in human embryos and highlight how human development can diverge from processes described in mouse.

**Decision letter after peer review:**

Thank you for submitting your article "Mechanism of cell polarisation and first lineage segregation in the human embryo" for consideration by *eLife*. Your article has been reviewed by 3 peer reviewers, one of whom is a member of our Board of Reviewing Editors, and the evaluation has been overseen by Kathryn Cheah as the Senior Editor. The reviewers have opted to remain anonymous.

The reviewers have discussed the reviews with one another and the Reviewing Editor has drafted this decision to help you prepare a revised submission.

The manuscript by Zhu and colleagues provides insight into the process of compaction and polarisation in human embryos. While a similar piece of work has already been published in Nature (Gerri et al., Nature 2019), all three reviewers were of the opinion that this work contains much information that should be published in *eLife* provided the experimental and interpretive issues described below can be addressed.

The main findings are the following:

• The authors report that the substantial part of the blastomere polarisation process is conserved between the mouse and humans (and possibly other mammals). Namely, the establishment and organisation of the apical (outside facing) domain follows two steps: changes in F-actin distributions, followed by Par complex localisation to the apical part of the blastomere.

• The process of polarisation is delayed in human embryos and is not dependent on compaction.

• Rearrangement of F-actin and assembly of apical Par complex requires Phospholipase C (PLC) signalling.

• Establishment of polarity is critical for GATA3 nuclear accumulation in the outside, polar cells. The study demonstrates the initiation of GATA3 expression in the trophectoderm is independent of the polarization process but is reinforced by it.

Essential revisions:

1. Reviewer 3 from the other journal mentioned the concerns about DMSO possibly affecting the experimental results. Given the problems they are having with DMSO treatment, they should prioritise the PLCE1+PLCB1 siRNA experiments, add some additional data and lead with these. The inhibitor data could then be used to support the siRNA and the caveats of the DMSO experiments discussed. Examples of the persistent problems with DMSO include Figure 2c and d, where it is clear that in both cases, a higher DMSO concentration results in a greater number of nonpolarized embryos, regardless of whether the embryos in question are or are not treated with an inhibitor. This appears the case for both F-actin and PARD6 polarisation, although statistics are applied to different comparisons in these two cases. In Figure 2f, it appears that the low dose of U73122 contains ehanced PARD6 apical enrichment compared to both concentrations of DMSO and the high concentration of inhibitor (and that this is significant), whereas the low dose of DMSO appears inhibitory. In addition, DMSO reduces GATA3 expression levels in Figure 3e. In general statistical tests should be applied to all treatments and the authors should discuss the caveats of these experiments in the text.

2. Improve the Imaging and interpretation of GATA3 expression, including DAPI staining to facilitate quantification. As it stands the authors data don't fully support their stipulation concerning GATA3 distribution. For example, in Figure 3c, normalised GATA3 intensity is much higher in non-polar (is this a typo?) than in polar cells. Or in Figure 3e, in embryos treated with high U3122 concentration, all expression all GATA3 expression is lost, not just the enhanced signal induced by polarisation. Similarly, some cells in the TE do not have any visible GATA3 staining. Can the authors provide information about the percentage of GATA3-positive and negative cells in both polar and apolar cells? Improved quantification and images, using DAPI to discern the nuclear signal, would benefit the manuscript and solidify any possible conclusions. Perhaps some of the siRNA embryos discussed above could also be used validate their observations about GATA3.

It is also difficult to understand the authors interpretation of the findings about GATA3 expression in mouse (the point also raised by Reviewer 3 in the letter from the other journal). Zhu and colleagues claim that GATA3 expression in human is different to that in mouse, where (as they claim) Gata3 expression is restricted to the outside cells. The authors wish to conclude that in human, this restriction happens later in development and that originally, GATA3 expressing cells are found at both inside and outside positions. Ralston et al., 2010 is used to support the claim authors claim about mouse Gata3 expression. However, in Ralston et al., 2010 GATA3 distribution is reported to be similar to CDX2 which is initially not restricted to the outside cells at the beginning of cell specification process. To be precise, Ralston and colleagues wrote: "Gata3 colocalized with Cdx2 in nuclei on a cell-by- cell basis (723/730 cells) in embryos examined at the 8- to 32-cell stages (31 embryos). Among embryos in which Gata3 and Cdx2 expression did not perfectly correlate (5/31 embryos), Gata3- positive/Cdx2-negative and Cdx2-positive/Gata3-negative nuclei were detected at equivalent frequency (four and three nuclei, respectively). Thus, Gata3 is coexpressed with Cdx2 from the earliest developmental stages ". With this in mind, the authors should revise their conclusions about the lack of evolutionary conservation between mouse and human. Alternatively, they are welcome to provide additional data to substantiate any claims.

3. Additional evidence blastomere internalization. While we appreciate the evidence based on relative polarity and the appearance of "larger" inside cells argues for internalization, they have only observed this once (out of 5 time lapse experiments). It would seem difficult to make a firm conclusion based on a single embryo and conclude "mechanistic difference of TE/ICM segregation between the two species."

In addition, the authors should be careful about their claims about the mouse here (as above). In the text they claim, "Therefore, whereas polarisation and subsequent asymmetric cell divisions are critical to generate inside cells in mouse embryos, in human embryos inside cells can be generated independently of polarity and asymmetric cell divisions, revealing mechanistic differences of TE/ICM segregation between the two species." It has been suggested in several recent publications that an asymmetric division may not be the only source of generating inside cells in mouse. Other processes like contractility-dependent cell internalisation may also be important (for example see Watanabe et al., 2014 or Anani et al., 2014).

4. The manuscript could make it more clear how the timing of compaction was established. Since in human embryos, compaction often starts asymmetrically (which often leads to one part of the embryo being already compacted, whereas other areas are not), how were the time points for the beginning, and endpoints of compaction demarcated? Did they use the first compacting cells to mark the start point and the last compacting cells the end point of the process, or were only some of the cells measured and others were discounted from the timing? There remains a lot of controversy in the field, regarding the timing and length of the compaction in human embryos (for example see Coticchio et al., 2019) and therefore the authors should moderate their claims.

5. The quantification of polarization in terms of hours post thawing is somewhat confusing, albeit important. The authors should also quantify cell polarisation relative to the total number of cells per embryo (for the data in Figure 1c-f). The representation of their data on polarity based on embryo cell number would help with the comparisons and an evaluation of the authors ideas about the temporal sequence of polarisation, compaction, and ICM formation. A similar conversion would be useful in Figure 3b.

[Editors' note: further revisions were suggested prior to acceptance, as described below.]

Thank you for resubmitting your work entitled "Human embryo polarization requires PLC signalling and mediates trophectoderm specification" for further consideration by *eLife*. Your revised article has been evaluated by Kathryn Cheah (Senior Editor), a Reviewing Editor and one of the original reviewers.

The manuscript has been improved but there are some remaining issues that need to be addressed, as outlined below:

1. The additional characterization of the RNAi experiments is a big improvement. However, DMSO continues to have a confounding effect, and it is very difficult to determine which comparisons have been performed – i.e. does U73122 have a more significant effect than any of the DMSO treatments? Is it only the higher concentration of DMSO that has a significant effect? Perhaps the sentence in the text could be further modified or, as the reviewer suggests the DMSO/U73122 experiments can either be removed or moved to supplemental.

2. GATA3 analysis is improved, but Figure 3c still shows the polarized cells expressing higher levels of GATA3 and PLC siRNA treatment seems to increase GATA3 intensity in polarized cells. Are these still labelling errors, or are there other issues with these experiments?

3. Please respond modify the text and/or respond to the two text issues raised by reviewer 2.

*Reviewer #2:*

In the revised manuscript Zhu and colleagues made a significant effort to address the reviewer's reservations. However, the main problems with the manuscript are still not appropriately addressed.

1. I still do not understand how the authors interpret the data on U73122 experiments. For F-actin polarisation, DMSO treatment has a stronger effect (in terms of the percentage of non-polarised embryos – and this I am guessing as the colour code is not included in the supplement figure 2a) plus in supplement figure 2b I cannot see if there is any difference between the appropriate DMSO control and the experimental group, and if there is any difference then there is a higher proportion of non-polarised blastomeres in the DMSO than in 7.5 μM U73122 group. At the same time, the authors say: "at a higher concentration (7.5 μM) U73122 significantly reduced the proportion of polarized blastomeres and the level of PARD6 apical enrichment". This statement seems to contradict what is reported in supplement figure 2a about F-actin staining as DMSO exerts a stronger effect than U73122 in supplement figure 2a (7.5 μM U73122 group and relevant DMSO control). As for the PARD6 polarisation (Figure 2c), the strongest effect can be seen in the 5 μM group where indeed U73122 has a stronger effect in comparison with the appropriate DMSO control but this is not so obvious in the 7.5 μM U73122 group. Admittedly, this is not very clear as the authors did not produce a statistical comparison between U73122 and an appropriate DMSO control. Nevertheless, the data are confusing and the statement in the discussion related to these data needs to be appropriately adjusted.

At the same time, the data from siRNA experiments are much clearer and more convincing, so again I urge the authors to move the emphasis to these data and remove some of the confusing data from inhibitor experiments.

2. Why is the average GATA3 intensity in polarised cells slightly higher after the PLC siRNA treatment than the control one in supplement Figure 3d? Is it a mistake in labelling or the data shows something different than what the authors suggested in the text?

3. In order to correctly represent the authors findings, the sentence: "Although we could only fluorescently label a small number of embryos to record the spatial localization and size of cells, our results suggest that cells are allocated to the ICM independently of the cell divisions of 8-cell stage blastomeres in the human embryo" should be revised to: Although we could only fluorescently label a small number of embryos to record the spatial localization and size of cells, our results suggest that at least some cells are allocated to the ICM independently of the cell divisions of 8-cell stage blastomeres in the human embryo.

4. Finally, I do not fully follow the authors argument that if human blastomeres can be potentially biased before compaction (which is as the authors stated independent from polarisation) that it means that the allocation to inner and outer positions has nothing to do with polarity. The fact that some blastomeres in some of the embryos were biased before compaction does not preclude the possibility that a group of originally "polarity" biased blastomeres are more prone to polarisation and then retain their outside position while "non-polar" biased blastomeres have a higher chance of becoming inside cells. At the moment, there is not enough data to argue either way. This part of the discussion has to be adjusted to appropriately report the findings from the authors and the papers the authors cite to support their claim.

---

## [Author Response]

Essential revisions:1. Reviewer 3 from the other journal mentioned the concerns about DMSO possibly affecting the experimental results. Given the problems they are having with DMSO treatment, they should prioritise the PLCE1+PLCB1 siRNA experiments, add some additional data and lead with these. The inhibitor data could then be used to support the siRNA and the caveats of the DMSO experiments discussed. Examples of the persistent problems with DMSO include Figure 2c and d, where it is clear that in both cases, a higher DMSO concentration results in a greater number of nonpolarized embryos, regardless of whether the embryos in question are or are not treated with an inhibitor. This appears the case for both F-actin and PARD6 polarisation, although statistics are applied to different comparisons in these two cases. In Figure 2f, it appears that the low dose of U73122 contains ehanced PARD6 apical enrichment compared to both concentrations of DMSO and the high concentration of inhibitor (and that this is significant), whereas the low dose of DMSO appears inhibitory. In addition, DMSO reduces GATA3 expression levels in Figure 3e. In general statistical tests should be applied to all treatments and the authors should discuss the caveats of these experiments in the text.

We agree with the reviewer that using RNAi is a more rigorous approach to test PLC’s function. This is why we carried out these experiments, despite that it was a challenge to carry them on human embryos (this delayed the publication of our results by nearly 2 years but it was important). We have now also performed additional characterisation of the PLC RNAi experiments, including analyses of the apical enrichment of PARD6 and F-actin (revised Figure 2h and Figure 2—figure supplement 1i), and GATA3 expression in relation to the proportion of polarised cells (revised Figure 3—figure supplement 1d).

We also now discuss the caveats of the PLC inhibitor experiments. Specifically, we say “While higher solvent carrier (DMSO) concentrations alone can have an impact on the number of polarized blastomeres, the addition of U73122 had a stronger effect.” (Page 7)

We have also performed statistical tests on F-actin and PARD6 polarisation and GATA3 expression in all treatments/conditions in the PLC inhibitor experiments, and have added the full range of statistical information of comparisons across all conditions in the figure legend. Our statistical analyses indicate that, although DMSO appears to have an effect on cell polarisation and GATA3 expression, U73122 7.5uM treatment significantly reduces apical PARD6 polarisation (Figure 2d) and GATA3 expression (Figure 3e) when comparing with either media control or vehicle control groups. These results suggest that U73122’s effect on polarisation and GATA3 expression is specific and is independent of that from DMSO.

2. Improve the Imaging and interpretation of GATA3 expression, including DAPI staining to facilitate quantification. As it stands the authors data don't fully support their stipulation concerning GATA3 distribution. For example, in Figure 3c, normalised GATA3 intensity is much higher in non-polar (is this a typo?) than in polar cells. Or in Figure 3e, in embryos treated with high U3122 concentration, all expression all GATA3 expression is lost, not just the enhanced signal induced by polarisation. Similarly, some cells in the TE do not have any visible GATA3 staining. Can the authors provide information about the percentage of GATA3-positive and negative cells in both polar and apolar cells? Improved quantification and images, using DAPI to discern the nuclear signal, would benefit the manuscript and solidify any possible conclusions. Perhaps some of the siRNA embryos discussed above could also be used validate their observations about GATA3.

We thank reviewer for the comment. The groups in Figure 3c were mis-labelled as the reviewer correctly pointed out. This has now been corrected. As the reviewer suggested, we have now quantified the percentage of GATA3 positive and negative cells in polar and apolar cells. This information is now provided in our revised Figure 3—figure supplement 1a. The DAPI channel for Figure 3 is now been provided in the updated Figure 3. While the DAPI signal is detected at a similar level in both inner and outer cells, there is a clear upregulation of GATA3 in outer cells. This supports our original conclusion about GATA3 expression pattern and its enrichment in polar cells as embryo develops.

It is also difficult to understand the authors interpretation of the findings about GATA3 expression in mouse (the point also raised by Reviewer 3 in the letter from the other journal). Zhu and colleagues claim that GATA3 expression in human is different to that in mouse, where (as they claim) Gata3 expression is restricted to the outside cells. The authors wish to conclude that in human, this restriction happens later in development and that originally, GATA3 expressing cells are found at both inside and outside positions. Ralston et al., 2010 is used to support the claim authors claim about mouse Gata3 expression. However, in Ralston et al., 2010 GATA3 distribution is reported to be similar to CDX2 which is initially not restricted to the outside cells at the beginning of cell specification process. To be precise, Ralston and colleagues wrote: "Gata3 colocalized with Cdx2 in nuclei on a cell-by- cell basis (723/730 cells) in embryos examined at the 8- to 32-cell stages (31 embryos). Among embryos in which Gata3 and Cdx2 expression did not perfectly correlate (5/31 embryos), Gata3- positive/Cdx2-negative and Cdx2-positive/Gata3-negative nuclei were detected at equivalent frequency (four and three nuclei, respectively). Thus, Gata3 is coexpressed with Cdx2 from the earliest developmental stages ". With this in mind, the authors should revise their conclusions about the lack of evolutionary conservation between mouse and human. Alternatively, they are welcome to provide additional data to substantiate any claims.

We thank the referee for this comment. In the original version of our manuscript, the claim that GATA3 expression was different between mouse and human embryo referred to the comparisons of GATA3 expression at the early blastocyst stage. In the mouse embryo at the early blastocyst stage, Gata3 signal is absent from inner cells (Ralston et al., 2010, Figure 4), whereas in human embryos a discernible GATA3 signal can be detected in inner cells as the embryo cavitates (our study, Figure 3a). We wanted to emphasize the late restriction of GATA3 expression in outer polar cells in the human embryo. We have clarified this point further in the manuscript by saying “We found that GATA3 was expressed in both polarized and non-polarized blastomeres even at the early blastocyst stage, although the nuclear signal intensity of GATA3 was significantly higher in polarized cells (Figure 3c and Figure 3—figure supplement 1a). This is in contrast to the mouse in which Gata3 expression become absent in the inner cells at the early blastocyst stage(Ralston et al., 2010).” (Page 5)

3. Additional evidence blastomere internalization. While we appreciate the evidence based on relative polarity and the appearance of "larger" inside cells argues for internalization, they have only observed this once (out of 5 time lapse experiments). It would seem difficult to make a firm conclusion based on a single embryo and conclude "mechanistic difference of TE/ICM segregation between the two species."

We agree with the reviewer that it would be beneficial to have more time-lapse experiments showing blastomere internalisation. However, this experiment is extremely challenging due to several factors. First of all, we pre-select embryos in which the compaction process is just commencing to increase the chances of visualising the internalisation event. Secondly, we do not have fluorescent transgenic reporters incorporated into human embryos and so we needed to establish an alternative way of labelling human embryos to follow cell fate. As the membrane dye we use internalises quickly, the membrane staining is not consistent across all embryos, and for some of them we cannot accurately determine individual cell positions. Therefore, to obtain a single good-quality for one embryo requires many additional embryos, which are extremely difficult to obtain. We admit in the manuscript that our conclusion for this specific question is based not only on lineage labelling but also on monitoring cell position and cell size, which together allowed us to conclude that cell allocation in human embryos is not dependent on cell divisions.

Excitingly, two papers were published during the revision of our manuscript that further support our conclusion. In Coorens et al., Nature, 2021 the analysis of somatic mutations in placental samples using whole-genome sequencing revealed that human blastomeres at the pre-compaction stage, and therefore before polarisation is established, have a biased lineage allocation. While Bizzotto et al., Science, 2021 confirmed such an early asymmetric contribution to extra-embryonic tissues by analysing somatic single nucleotide variants. Globally, these two landmark studies together with our time-lapse observation indicate that the asymmetric allocation of the apical domain is not the only path to generate inside cells. We have added this information in our revised discussion. Specifically, we say “In further support of this hypothesis, two recent studies have demonstrated that blastomeres at the pre-compaction stage, and therefore before polarization is established, have a biased lineage allocation (Bizzotto et al., 2021; Custers et al., 2021) ”. (Page 7)

In addition, the authors should be careful about their claims about the mouse here (as above). In the text they claim, "Therefore, whereas polarisation and subsequent asymmetric cell divisions are critical to generate inside cells in mouse embryos, in human embryos inside cells can be generated independently of polarity and asymmetric cell divisions, revealing mechanistic differences of TE/ICM segregation between the two species." It has been suggested in several recent publications that an asymmetric division may not be the only source of generating inside cells in mouse. Other processes like contractility-dependent cell internalisation may also be important (for example see Watanabe et al., 2014 or Anani et al., 2014).

We are grateful for this comment and would like to clarify that when we said “asymmetric cell division” we were referring to divisions that happen after embryo compaction, in which the apical polarity domain becomes asymmetrically inherited by the two daughter cells, resulting in one polar and one apolar cell. Anani et al., 2014 and Maitre et al., 2016 suggest that the contractility-dependent cell internalisation is an event that is downstream of this polarity divergence. Our live imaging experiment suggest that in human embryo, the inner position of the cell can be obtained during the process of compaction. The timing difference of when inner cells are generated (compacting stage in human vs. post-compaction stage) indicate a different mechanism involved in human embryo to generate inner cells, and which we suggest that the physical restriction of the compaction process as a possible route (“Discussion” section, page 7). We agree with the reviewer that our current dataset cannot rule out the possibility that contractility-dependent mechanism may also be involved, and therefore we revised our statements by now saying: “Although we could only fluorescently label a small number of embryos to record the spatial localization and size of cells, our results suggest that cells are allocated to the ICM independently of the cell divisions of 8-cell stage blastomeres in the human embryo.”

4. The manuscript could make it more clear how the timing of compaction was established. Since in human embryos, compaction often starts asymmetrically (which often leads to one part of the embryo being already compacted, whereas other areas are not), how were the time points for the beginning, and endpoints of compaction demarcated? Did they use the first compacting cells to mark the start point and the last compacting cells the end point of the process, or were only some of the cells measured and others were discounted from the timing? There remains a lot of controversy in the field, regarding the timing and length of the compaction in human embryos (for example see Coticchio et al., 2019) and therefore the authors should moderate their claims.

We thank the reviewer for this comment – it was indeed important for us to clarify the criteria we have used to evaluate compaction. As indicated in the revised version of the manuscript (page 3) “Based on previously established criteria (Zhu, Leung, Shahbazi, and Zernicka-Goetz, 2017), we defined the initiation of compaction as the time at which the inter-blastomere angle increased to greater than 120° between any two blastomeres, and the completion of compaction as the time at which all angles were greater than 120°.”(Page 18). This precise analysis of compaction allow us to confidently determine the timing of compaction in the human embryo.

5. The quantification of polarization in terms of hours post thawing is somewhat confusing, albeit important. The authors should also quantify cell polarisation relative to the total number of cells per embryo (for the data in Figure 1c-f). The representation of their data on polarity based on embryo cell number would help with the comparisons and an evaluation of the authors ideas about the temporal sequence of polarisation, compaction, and ICM formation. A similar conversion would be useful in Figure 3b.

We thank for this advice. We now provide the analysis of the number of polarised cells as a function of the total number of cells in each embryo (revised Figure 1—figure supplement 1j). We found a significant positive correlation between the total number of cells and the number of polarised cells (r = 0.8550, p<0.0001). This data complements our previous analysis on cell numbers and percentage of polarised cells over time.

[Editors' note: further revisions were suggested prior to acceptance, as described below.]

The manuscript has been improved but there are some remaining issues that need to be addressed, as outlined below:1. The additional characterization of the RNAi experiments is a big improvement. However, DMSO continues to have a confounding effect, and it is very difficult to determine which comparisons have been performed – i.e. does U73122 have a more significant effect than any of the DMSO treatments? Is it only the higher concentration of DMSO that has a significant effect? Perhaps the sentence in the text could be further modified or, as the reviewer suggests the DMSO/U73122 experiments can either be removed or moved to supplemental.

We thank the reviewers and editor for this comment. We agree that DMSO treatment itself also causes a polarity phenotype, and therefore this makes the PLC inhibitor experiments non-conclusive. For this reason, in the revised manuscript, we have decided to move all DMSO/U73122 experiments into supplementary. Moreover, we acknowledge the limitation of this experiment in the Results section. In the revised version of the manuscript, we have also included the statistical significance of all pairwise comparisons in the figure legend (including these results in the graph itself would make the figure very cluttered).

2. GATA3 analysis is improved, but Figure 3c still shows the polarized cells expressing higher levels of GATA3 and PLC siRNA treatment seems to increase GATA3 intensity in polarized cells. Are these still labelling errors, or are there other issues with these experiments?

The labels of Figure 3c were swapped. We apologize for this mistake, which we have now corrected. GATA3 levels are higher in polarized cells compared to non-polarized cells as shown in the corrected Figure 3c.

Regarding the PLC siRNA experiment, we have now compared the levels of GATA3 in polarized control cells to the levels of GATA3 in polarized PLC siRNA-treated cells. The statistical result shows that GATA3 levels are not significantly different in polarized cells of control and PLC siRNA treated embryos (indicated in the legend of Figure 3 supplement 1f). This indicates that only in cells that fail to polarize, GATA3 levels are significantly decreased. We have also clarified this point in the Results section of the manuscript.

3. Please respond modify the text and/or respond to the two text issues raised by reviewer 2.

We have revised the text in response to all relevant comments of reviewer 2.

Reviewer #2:In the revised manuscript Zhu and colleagues made a significant effort to address the reviewer's reservations. However, the main problems with the manuscript are still not appropriately addressed.1. I still do not understand how the authors interpret the data on U73122 experiments. For F-actin polarisation, DMSO treatment has a stronger effect (in terms of the percentage of non-polarised embryos – and this I am guessing as the colour code is not included in the supplement figure 2a) plus in supplement figure 2b I cannot see if there is any difference between the appropriate DMSO control and the experimental group, and if there is any difference then there is a higher proportion of non-polarised blastomeres in the DMSO than in 7.5 μM U73122 group. At the same time, the authors say: "at a higher concentration (7.5 μM) U73122 significantly reduced the proportion of polarized blastomeres and the level of PARD6 apical enrichment". This statement seems to contradict what is reported in supplement figure 2a about F-actin staining as DMSO exerts a stronger effect than U73122 in supplement figure 2a (7.5 μM U73122 group and relevant DMSO control). As for the PARD6 polarisation (Figure 2c), the strongest effect can be seen in the 5 μM group where indeed U73122 has a stronger effect in comparison with the appropriate DMSO control but this is not so obvious in the 7.5 μM U73122 group. Admittedly, this is not very clear as the authors did not produce a statistical comparison between U73122 and an appropriate DMSO control. Nevertheless, the data are confusing and the statement in the discussion related to these data needs to be appropriately adjusted.At the same time, the data from siRNA experiments are much clearer and more convincing, so again I urge the authors to move the emphasis to these data and remove some of the confusing data from inhibitor experiments.

We thank the reviewer for the comment and apologize for the omission of the legend. We agree that DMSO treatment itself causes a polarity phenotype, and therefore this makes the PLC inhibitor experiments non-conclusive. For this reason, in the revised manuscript, we have decided to move all DMSO/U73122 experiments into supplementary. Moreover, we acknowledge the limitation of this experiment in the Results section.

We have also decided to remove the panels that presented a non-quantitative analysis of embryo polarization, which based on the reviewer’s comment were not entirely clear. We are only including the apical/cytoplasmic ratio of F-actin and Par6 as a quantitative measure of polarization. All the pair-wise statistical comparisons for Figure 2 supplement 1c and 1d are indicated in the legend. While the comparison medium control vs. PLC inhibitor is statistically significant for 7.5uM (high concentration), the comparisons between PLC inhibitor treatment and DMSO control are not significant. This is now clearly specified in the Results section. Given the confounding effect of DMSO in terms of polarization, we have focused on the siRNA treatment experiment as suggested by both reviewers.

2. Why is the average GATA3 intensity in polarised cells slightly higher after the PLC siRNA treatment than the control one in supplement Figure 3d? Is it a mistake in labelling or the data shows something different than what the authors suggested in the text?

We thank the reviewer for the careful observation. We have compared the levels of GATA3 in polarized control cells to the levels of GATA3 in polarized PLC siRNA-treated cells. The statistical result shows that GATA3 levels are not significantly different in polarized cells of control and PLC siRNA treated embryos (indicated in the legend of Figure 3 supplement 1f). This indicates that only in cells that fail to polarize, GATA3 levels are significantly decreased. In other words, the effect of PLC siRNA on GATA3 expression is through a reduction of the number of polarized cells. We have also clarified this point in the Results section of the manuscript.

3. In order to correctly represent the authors findings, the sentence: "Although we could only fluorescently label a small number of embryos to record the spatial localization and size of cells, our results suggest that cells are allocated to the ICM independently of the cell divisions of 8-cell stage blastomeres in the human embryo" should be revised to: Although we could only fluorescently label a small number of embryos to record the spatial localization and size of cells, our results suggest that at least some cells are allocated to the ICM independently of the cell divisions of 8-cell stage blastomeres in the human embryo.

We thank the reviewer for the helpful suggestion and have modified the text as the reviewer suggested.

4. Finally, I do not fully follow the authors argument that if human blastomeres can be potentially biased before compaction (which is as the authors stated independent from polarisation) that it means that the allocation to inner and outer positions has nothing to do with polarity. The fact that some blastomeres in some of the embryos were biased before compaction does not preclude the possibility that a group of originally "polarity" biased blastomeres are more prone to polarisation and then retain their outside position while "non-polar" biased blastomeres have a higher chance of becoming inside cells. At the moment, there is not enough data to argue either way. This part of the discussion has to be adjusted to appropriately report the findings from the authors and the papers the authors cite to support their claim.

We agree with the reviewer that the data from the two new studies are indirect and cannot preclude the possibility of the involvement of polarisation. We have therefore adjusted our discussion by now saying: “This early asymmetric contribution to either the embryonic or the extra-embryonic tissues suggests the possibility that some inner cells may be generated independently of embryo polarization at the 8-cell stage, as our observation here”.

[Editors' note: we include below the reviews that the authors received from another journal, along with the authors’ responses.]

Referee #1 (Remarks to the Author):The manuscript by Zhu et al. shows that human embryos undergo compaction followed by cell polarization, and that PLC activity is required for polarization to occur. Interestingly, the authors suggest that initiation of lineage specification is independent of cell polarization. The study is based on morphokinetic and immunofluorescence (IF) analysis of vitrified-warmed IVF derived human embryos. Based on timing of embryo compaction and localization of Factin and PARD6, the authors conclude that polarization happens in two phases, with polarization of F-actin concomitant with compaction followed by polarization of PARD6 (which is similar to the events of mouse embryo development). Using a PLC inhibitor during human embryo culture, the authors tested the hypothesis that PLC activity is required for polarization. From their experiments they concluded that, as in mouse, PLC activity is required for polarization and since compaction was not considered to be regulated by PLC activity, they conclude that compaction and polarization are independently triggered. Finally, the authors stained embryos for GATA3 expression in control and PLC inhibited embryos, and found that initiation of GATA3 expression is independent of cell polarization, while polarization was associated with increased levels of GATA3 expression.While the data presented is interesting and novel, many questions still remain and some conclusions are not fully supported by experimental results or need further validation. Also, most of the results are a confirmation of what is known in mice, and do not offer an improved mechanistic understanding of early mammalian development. The main exception being that human embryos seem to activate GATA3 independent of cell polarization, but this conclusion is not strongly supported and is not sufficiently explored.

We thank this reviewer for the supportive summary of our work and his/her appreciation of the novelty of the data. While several of our findings confirm observations in mouse embryos, as reviewer 3 points out it is both “interesting and essential to repeat on humans the studies performed on the mouse, even if the conclusions are very similar”.

Thus, overall, while the characterization of human IVF embryo development is useful and novel, it does not raise to the level of novelty and significance that I would consider sufficient for a publication in our journal. Also, the letter format that the authors pursue, negate sufficient space for discussing the findings and putting them in context of the current literature, for example, how this work relates to human embryo single-cell gene expression data (Petropoulus et al., PMID: 27062923) implying that TE, EPI and PE emerge all at the same time at the blastocyst stage.

Our manuscript presents the first characterisation of human embryo polarisation and trophectoderm-inner cell mass segregation, and the upstream mechanism that tiggers this process, and therefore in our opinion it raises to the level of novelty required for publication. While (Petropoulos et al., 2016) describe that TE, EPI and PE lineages become specified at the blastocyst stage, they also show that the initiation of TE marker expression begins at the compacted morula stage. At this stage, blastomeres co-express TE genes such as GATA3 and PDGFA with ICM markers. Therefore, our immunofluorescence results are in agreement with the single cell sequencing expression data of Petropoulos et al. We mention this in the revised version of the manuscript on the page 5. We say “…, we analysed the spatiotemporal expression of the transcription factor GATA3, which is determinant for TE specification and based on single-cell RNA sequencing is already induced at the compacting morula stage” (page 5).

Major issues:While the authors show that PLC activity may be required for cell polarization, many mechanistic questions remain, e.g. what activates PLC? How PLC induces Actin restructuring? How is GATA3 activated and how is it later restricted to outside cells? These all seem to be important mechanistic questions not answered in the current report.

These are very important mechanistic questions, and for most of them we still do not have a clear answer even in the much less demanding experimental system of the mouse embryo. Addressing these questions would require a vast number of cleavage-stage human embryos, which would have to be created for research. This would be regarded as unethical and indeed, under our current licences this is not possible. To characterise the dynamics of polarisation and the function of PLC we have already used almost 300 human embryos, which is unprecedented. To address all the questions that the referee mentions, we would need at least an additional thousand human embryos to be able to test the different hypotheses. This is simply impossible.

Embryos in Figure 2 seem to be at different levels of compaction, and thus different levels of polarization. Also, in the figure the control embryo has significantly more cells than treated embryos. Authors should change the image for embryos that are all at the same level of compaction and with similar cell number. Otherwise, treated embryos seem to undergo a developmental delay, and based on figure 1, PARD6 polarization would not be expected for uncompacted embryos. There is no comment of figure S2D, which shows a substantial difference in compaction between controls and all treatments, including DMSO, but specifically for 7.5 μm treatment with only 10% embryos undergoing compaction. Also, this data doesn’t have statistical analysis. Given the effect of DMSO and U73122 on compaction, it seems that it may be necessary to evaluate polarization in compacted embryos/cells, to be able to conclude that polarization and compaction are independent of each other with only polarization being sensitive to PLC inhibition; as opposed to looking at the two aspects individually.

We thank reviewer for the comments. We have replaced the representative images shown in Figure 2b to better reflect the quantification shown in Figure 2c-d. As shown in Figure 2e, there are no significant differences in the number of cells when comparing control embryos, embryos treated with DMSO and embryos treated with U73122. This indicates that the cell polarisation defects in the PLC treatment group are not a result of developmental delay.

To respond to the referee’s comment, we have repeated the PLC inhibitor experiment to increase the number of embryos analysed. As we show in the new Extended Data Figure 3d there are no significant differences in the rate of compaction between control embryos and DMSO-treated embryos (both at 5 and 7.5 µM). Embryos treated with the PLC inhibitor at 7.5 µM present a mild reduction in compaction. Therefore, we cannot exclude that the PLC pathways contribute to the process of compaction. We have modified our conclusions accordingly and now state “These results indicate that PLC is a major regulator of cell polarisation, but not of compaction.” (page 4).

Authors need to acknowledge that data was generated from IVF, vitrified/thawed embryos provided by couples seeking infertility treatment, and therefore exact timing of events may not be representative of “normal embryo” development. It would be valuable for the authors to compare analysis of fresh versus vitrified/thawed embryos (or those vitrified at the 2PN versus the 8-cell stage) to show whether or not the vitrified/thawed embryos are representative.

We thank the reviewer for the comment and acknowledge that data was generated from IVF embryos provided by couples seeking infertility treatment, and therefore descriptions of preimplantation events may not be representative of “normal embryo” development. However, since the donated surplus embryos of this study were obtained from young women (mean age 28.12±4 years; N=260), we can anticipate some correlation to “normal embryo” development. Although the potential impact of a severe male-factor aetiology on embryo development cannot be ruled out, maternal age is one of the strongest predictors of human embryo competence because of the increased incidence of meiotic aneuploidy or other age-related factors that contribute to reproductive senescence (Reig et al., 2020). Moreover, all the day-3 embryos used in our study were considered suitable for embryo freezing and hence for subsequent cryotransfer. The average cell number at the time of embryo warming was 7.98±1.26 (N=260), which is consistent with the 8 cells expected on day-3 stage of development (Α Scientists in Reproductive and Embryology, 2011). Please see the revised methods section for details.

We also acknowledge that data was generated from vitrified/thawed embryos and most vitrification times were unknown, therefore embryos potentially having different baseline vitrification times may skew the reported timings. Please see the revised methods section. The expected stage of embryo development at each specific time point has been previously described (Α Scientists in Reproductive and Embryology, 2011), and the morula stage is expected approximately at 92±2 hours post fertilisation (h.p.f). We analysed compacting embryos in which the exact times of fertilisation and cryopreservation were known, and the completion of compaction was accomplished at 80.2±8.5 h.p.f (new Extended Data Figure 1j).

We now specify this in the text (page 3).

Similarly, we analysed immunostained embryos in which the exact times of fertilisation and cryopreservation were known. Considering that the day 3 developmental stage extends approximately from 67-90 h.p.f (Α Scientists in Reproductive and Embryology, 2011), we divided these embryos into three groups: 74-75 h.p.f (early day 3, represents 2h fixation time), 77-80 hpf (mid-day 3, represents 9h fixation time) and 93-97 hpf (day 4, represents 24h fixation time). At early day 3, none of the embryos had commenced polarisation (no cells polarised for F-actin or Pard6). At mid-day 3 some cells started to show F-actin polarisation but no cells were yet polarised for Pard6. At day 4, Pard6 polarisation occurred, antagonising F-actin polarisation, as reflected in 41 Pard6 polarised cells vs. 21 F-actin polarised cells at 93-97 h.p.f. Therefore, we have observed how human embryo polarisation is established sequentially during days 3-4 of development, with F-actin polarisation preceding Pard6 polarisation (please see new Figure 1e, f).

Please see that in Figure 2. (c), numbers indicating cells analysed are inverted between the categories.

We thank the reviewer for pointing this out. This has been corrected.

Likewise, there are some live analyses (i.e. lifeACT, and the expression of fluorescently tagged transcripts) that could be valuable even in a limited number of embryos alongside the current robust population analysis.

We thank the reviewer for this comment and agree that live-imaging would be beneficial to strengthen the conclusions in our manuscript. To this end, we have performed a new experiment using live imaging to follow cell positions during human embryo development. Specifically, we applied a cell-membrane live dye (FM 4-64FX) to label cell membranes, and examined cell dynamics during the compaction process. These data are included in revised Extended Figure 4d. In agreement with our conclusion in the original version of this manuscript, the 5 embryos that we imaged begin to compact from 8-cell stage onwards (new Extended Figure 4d). More importantly, we found that in one out of the 5 embryos that we examined in this detailed way, one inner cell had been generated during the compaction process (new Extended Figure 4d). To our knowledge this is the first fluorescence live imaging that has ever been done on the human embryo. This new experiment significantly strengthens our conclusions about compaction timing and the differential mechanisms of inner cell generation in the human embryo.

It is not clear how the inside/outside cell assessment in the last paragraph of the manuscript relates to the rest of the report, especially, what is the relevance of the conclusion that human embryos may have inner cells before asymmetric cell divisions start?

We thank the reviewer for the comment and apologise for the lack of clarity. In the mouse embryo, the inside positioning of the cell is crucial for ICM fate specification. For this reason, we analysed whether the same mechanism is conserved in human embryos. In the mouse embryo, inner cells are first generated by so-called asymmetric cell divisions at the 8-16 cell stage transition. The observation that inner cells can be generated before such cell divisions are initiated in human embryos, suggests mechanistic differences of TE/ICM segregation between mouse and human embryos. We have now clarified this in the text. We say “Therefore, whereas polarisation and subsequent asymmetric cell divisions are critical to generate inside cells in mouse embryos, inside cells can be generated independently of polarity and asymmetric divisions in human embryos, revealing mechanistic differences of TE/ICM segregation between the two species.” (page 6)

Abstract conclusions and manuscript conclusions contradict each other.

We apologise for the lack of clarity. We have corrected this in the revised manuscript.

Minor issues:Sometimes authors refer to cells and sometimes to embryos, this is very confusing, whenever referring to cells, the authors should indicate the number of embryos represented. E.g., 3rd text paragraph discussing compacted cells/embryos.

We thank reviewer for the comment. We have added information on the number of embryos in the relevant place in figure legends.

How and which angle between cells was measured is unclear. How was the angle between cells measured for cell on top/bottom of embryos? A diagram and some examples would be helpful. Additionally, it is unclear how the angle of 120O was chosen as the definition of compaction.

We apologise for the lack of clarity. We have indicated in Extended Figure 1g which angle was measured to determine the degree of embryo compaction. In addition, we have included a detailed description of how this measurement was performed in the methods section. We agree with the reviewer that selecting 120° as a cut-off is arbitrary, but in our experience it has allowed us to differentiate compacted and non-compacted embryos as shown in (Zhu et al., 2017).

How inside and outside cells were defined is not clear. Please provide 3D images of inside cells, is it possible that cells with GATA3 expression may have an apical domain facing the top/bottom of embryo, which is not captured by the 2D images shown?

We have determined cell position by examining the cell area localising outside of the embryo. The edge of the embryo was set on both XY and along the Z axis by looking at F-actin localization, and the area of the cell of interest was examined on each plane. Cells that had an area exposed to the outside were defined as outer cells, whereas cells that did not have an area exposed to the outside were defined as inner cells. We have clarified this in the methods section (page 17-18). In addition, we now provide a 3D reconstruction of the embryo in the new Figure 4c and Extended Figure 4d.

How long was morphokinetic analysis carried for? What was the blastocyst formation rate for vitrified/thawed embryos? This is a better indication of embryo quality than morphological assessment of cell-viability.

The morphokinetic analysis was performed from the start of the experiment immediately after embryo warming (Day-3 vitrified embryos) until embryo fixation at 2, 9 or 24 hours after warming. We have now clarified this in the methods section (page 17). In our work we focused on days 3 and 4 of development, since it is exactly when the events of compaction and polarisation take place. Embryos were fixed at these time points and therefore, it was not possible to determine the blastocyst formation rate.

How were embryos generated? ICSI? What was the reason for IVF/ICSI of couples providing embryos? Is there any difference in reason for IVF in young vs advanced maternal age patients that may explain the difference in time of compaction? Otherwise, can the authors propose any reason for this difference. Further breakdown of embryos acquired from couples undergoing fertility treatment versus the 15 donated embryos would help indicate whether or not some of these features are artifacts of unhealthy embryos or representative of healthy embryos.

The majority of the day-3 embryos used in this study were generated by ICSI (76.2%=198/260) using donor eggs (72.7%=189/260). The average age of patients providing oocytes was 28.12±4 years (N=260), which is one of the strongest predictors of embryonic competence and oocyte quality (Cimadomo et al., 2018). The mean age of male patients providing sperm was 40.8 ± 6.98 (N=260). Most common female indications for ART included age (55%=143/260), poor ovarian response (8.1%=21/260) and tubal factor (7.7%=20/260). We have now included these details in the methods section of our manuscript (page 13).

Our observation of a negative correlation between maternal age and compaction time was surprising. Since our initial dataset did not include patients of advanced maternal age, we decided to analyse a larger cohort of embryos – 810 – which included mothers of very different ages, from less than 20 to more than 40 years old. These additional analyses revealed a lack of a significant correlation between these two variables (see Author response image 1). We have removed this analysis from the manuscript.

**Author response image 1. sa2fig1:** Correlation analysis of the variables ‘duration of compaction’ and ‘female age’ in a validation dataset comprising 810 embryos. Duration of compaction was defined as the time in hours between the start of compaction and morula formation. r: pearson correlation coefficient; p: p-value. No correlation was found between both variables..

The writing at times lacks precisions with the use of some terms/concepts: -Abstract: “totipotent egg” is inappropriate. It could be totipotent zygote, or totipotent fertilized egg, but the egg (MII stage oocyte) itself is not considered totipotent.-Abstract: the trophectoderm gives rise to the trophoblast of the placenta, while the connective tissue of the placenta (mesenchyme) is derived from ICM-derived mesoderm. So, if the authors want to be precise, the ICM also contributes to placenta development.-abstract indicates that manuscript provides “transcriptional events…” this is not true, as only protein presence/localization is shown, not transcriptional activity.-Text: when referring to vitrified embryos, thawing doesn’t exist (as there is no freezing), so authors should refer to “warming” of vitrified embryos, not “thawing” vitrified embryos. This is correctly described in the methods, but often misused in the main text.-Methods: primary antibodies used are not anti-Rabbit or anti-Mouse, they are likely raised in rabbit or mouse, against human/mouse proteins. Please correct and clarify. Also, Phalloidin is not an antibody.

We thank the reviewer for carefully reading our manuscript. We very much appreciate this and have corrected these errors in the revised version of our manuscript.

Statistical analysis are not described for most comparisons done. When cells are considered the experimental unit, was the data blocked for each embryo? This may be important especially when comparing fluorescence intensity.

We apologise for this omission. We now indicate the statistical test used for each experiment in figure legends. A summary of the statistical tests used is also provided in the methods section. We have re-analysed the percentage of polarised embryos using the embryo, instead of the cell, as the experimental unit. To do this we have calculated the percentage of polarised cells per embryo, and we have binned the data in 3 different groups. This analysis at the embryo-level also revealed significant differences between the DMSO-treated and PLCitreated embryos (new Figures 2c-d). For the quantification of the levels of fluorescence intensity we have not performed a blocked analysis, as the polarisation of a cell in an embryo is independent from the rest of the cells (Johnson and Ziomek, 1981).

Amount of DMSO used in controls should be indicated.

We apologise for this omission. In the revised version of the manuscript we include the specific amounts of DMSO used in controls.

Reference 7 and 17 are the same.

We apologise for the mistake. This has now been corrected.

Figure 1: Is the PB always PARD6 positive?

We observe that the PARD6 antibody shows a none-to-weak cytoplasmic signal on the polar body (Figure 2b). The bright spot shown on Figure 1b labels the lysing polar body. Cells that are undergoing cell death can give a non-specific antibody signal, which may be happening in this case.

Panels C and D: was F-actin and PARD6 polarization only measured in a single focal plane per embryo? Panels e and f need a trend line. Ideally, a correlation analysis should be done and authors should indicate the R and P values.

F-actin and PARD6 polarisation were measured in a single focal plane, by taking the middle plane of the embryo. We calculated the relative apical F-actin and PARD6 membrane enrichment for each cell. We now clarify this in the methods section (pages17-18). We have also added the correlation analysis by adding the trend line and R and P values as the reviewer suggested.

Figure 2: Panel a: the groups indicated are not clear. What does it mean group 2: 5uM; 7.5uM DMSO? Why separate in groups, if each was individually evaluated? Was DMSO added at 5 and 7.5 μm or to the same level used in U73122 treatments?

We apologize for the confusion: for U73122 treatment we assessed two concentrations, 5 and 7.5 uM. For each concentration we designed the corresponding control group in which we applied DMSO (the carrier of U73122) to the media at the same concentration. We have now modified the figure based on the reviewer’s comment.

Figure 3: the number of embryos evaluated per group is not clear. Can n=X be added for each experimental group?

This information was already present in the figure legends. We have now included this information also in the Results section.

Figure 4: is it possible that inner cell in cross section has an apical domain towards top or bottom of embryo (not captured by 2D image)? Furthermore, it should be stated for each image if it is a single z-plane or a full or partial projection of the embryo.

To address this question, we now provide a 3D reconstruction image of the inner cells we presented in Figure 3a and Figure 4a, b. This 3D reconstruction shows the inner position of the cells (please see new Figure 4c). In addition, we specify that the images in Figure 3a and Figure 4a, b are single planes.

Figure S1: panel G: How was hpf calculated if embryo freezing time was not available? Panel I and j: please add trend/correlation line and indicate the r value.

Fertilisation and vitrification times were available in 40/260=15.4% of embryos. We used one of these embryos for this figure. Panels S1i and S1j have been removed in the revised version of the manuscript, as the correlations were not significant for the new embryos analysed.

Figure S2: Panel b needs Y-axis label and all mean lines should be same color (red or black). Panel d needs statistics.

These corrections have been made in the revised version of the manuscript.

Referee #2 (Remarks to the Author):This manuscript reports the preimplantation development of human embryos. The authors used 235 surplus human embryos to analyze the first cell lineage segregation. They focused on compaction and polarization and found that these morphogenetic events happen similar to mouse development, prior to the expression of a lineage specific marker, GATA3. They conclude that the first lineage segregation in human preimplantation embryos is regulated by polarization similar to mouse embryos. The results in this manuscript are interesting and important, however descriptive at the current form.

We thank this reviewer for the supportive summary of our work and her/his appreciation of the novelty of the data. We would like to indicate that we now provide two types of functional experiments that demonstrate the essential role of the PLC pathway in inducing polarisation of the pre-implantation human embryo.

Minor commentsThe authors stated that a negative correlation of the compacting period with maternal age. This is interesting. However, I have a concern about sampling bias. This should be discussed.

We thank the reviewer for this insightful suggestion. As the reviewer points out, our dataset did not include cases of advanced maternal age. For this reason, we have repeated this analysis using a larger cohort of embryos (810), which includes mothers of very different age and did not detect a significant correlation between these two variables and consequently removed this statement from the manuscript.

Figure 3a Example 2, GATA3 staining. The image size is different from others.

We apologize and we have corrected this. In the revised version of the manuscript all images from the same embryo have the same magnification.

Supplementary Figure 1Figure 1a the numbers in X-axis is not aligned with the bars in the graph. Figure 1b the data looks binned. But not clear how the authors did.

This has now been corrected. The data shown in S1b was automatically binned using a statistics analysis software and it is presented as a histogram.

Referee #3 (Remarks to the Author):In “Mechanism of cell polarization and first lineage segregation in the human embryo”, Zhu, Shahbazi, Martin and colleagues investigate the timing and molecular control of human preimplantation embryo cell polarization and lineage specification. This study tackles a fundamental aspect of human development building upon decades of studies on the mouse embryo by different groups, including essential contributions of the lab of Prof Zernicka-Goetz. Using time-lapse microscopy of human embryos, chemical inhibition of PLC and immunostaining, they characterize the timing of polarization and TE differentiation of human blastomeres and identify that, like in the mouse, PLC activity is required. The manuscript is clearly written with and interesting narrative. This narrative is not always supported by data from the literature or from the present study. On the contrary, the data and the literature sometimes contradict the narrative proposed by the authors.The question is simple (and nevertheless essential) and the few simple experiments are directly addressing it. The experimental and analytical methods are not sufficiently explained in the methods or figure legends to be understood clearly or to be replicated.The figures are clear. However, the way the data are reported can be misleading with omission of critical statistical tests and misrepresentation, for example of cell instead of embryo or experiment replicate number. Finally, there is a potential critical experimental flaw, which is not addressed by the authors.Together, I believe the conclusions of the authors are mostly correct and I would probably only marginally disagree with them. In my opinion, the authors are mostly correct because the conclusion is that human preimplantation embryos behave mostly like the mouse ones. It still think it is interesting and essential to repeat on humans the studies performed on the mouse, even if the conclusion are very similar. However, these experiments need to be performed and reported rigorously.

We thank this reviewer for this supportive summary of our work and his appreciation of the novelty and significance of the data. We have addressed the criticism of the reviewer and our detailed responses are below.

Major issues:1) I suspect a potential critical experimental flaw with the chemical inhibition. Figure S2d, reports that most embryos in DMSO show developmental defects as compared to control embryos (6/40 dead, 19/40 fail to compact). I find very unfortunate that the authors did not comment on this and omitted comparing the control and DMSO groups during their statistical tests shown in Figure 2, 3 and S2 (especially when looking at the difference between control and DMSO groups in Fig2d). Instead, the authors simply state “without an obvious effect on cell cycle length (Figure 2b-f and Extended Data Figure 2a, b)” or “it did not have a significant effect on embryo compaction compared to the DMSO control group (Extended Data Figure 2d)”, which is technically correct but depict a biased view of the facts (see my later comment regarding the authors narrative on that). When looking for how much DMSO was used, I could not find the information (see my other major issue regarding methods). All I could find is indicated on the schematic of Figure 2 “Group 2: 5 µM; 7.5 µM DMSO”, which I assume is not telling that 5 µM DMSO were used but instead stands as an incorrect way of indicating that the DMSO concentration corresponds to the one used to dilute the PLC inhibitor to 5 and 7.5 µM respectively. It is essential to indicate how much DMSO is used and, if it causes so many developmental problems, it is probably too much to conclude safely. Therefore, these experiments do not support the narrative of the authors. Different approaches could be used (another inhibitor, lower concentrations, RNAi, CRISPR, …) but I will let the authors and their ethical committee decide whether they are worth it. In any case, the data of the study cannot be published without at the very least commenting on this particular effect of DMSO and it would be of highest honesty to show the panel of Figure S2 in the main figure.

We thank reviewer for the comment. As for the DMSO’s effect on embryo development, the effect we saw was likely due to a small sample size. To examine this possibility, we performed additional repeats for DMSO and U73122 treatments to enlarge our sample size. With the combination of new experimental results, we cannot detect a significant effect of DMSO compared to the controls. Our data indicates that the effect of PLC inhibition on cell polarisation is specific to this experimental condition.

To further verify the role of PLC in cell polarity establishment in the human embryo, we have now conducted experiments to use the RNAi approach to deplete two highly expressed PLC isoforms (new Figure 2g-i; new Extended Data Figure 3e-f). Our new results show that the depletion of PLCE1 and PLCB1 leads to a reduced number of polarised cells, phenocopying the results of inhibitor treatment. Both of these independent functional experiments indicate that PLC regulates cell polarisation in the early human embryo.

2) Methods and figure legends do not give sufficient information and, although that may not necessarily change the conclusions of the authors, I suspect that some of the statistical tests are incorrectly used.

We apologise for not providing all the necessary information. We have now added statistical test information for all experiments in the paper.

Experimental methods do not indicate how much DMSO or antibody are used. What types of illumination and image acquisition are used? What objectively defines a cell as polarized vs non-polarized, GATA3 positive vs negative? How are measured the length or intensity of the apical/basal domains, area of cells or GATA3 levels. Are these measurement performed on 2D images or on 3D volumes? How are the confocal slices chosen? This is all insufficiently described in the methods and legends. I would encourage the authors to provide the raw data, ROIs and numbers.

Our original submission already included a description of the type of microscope SP5 confocal microscope (Leica Microsystems) and objective used (40x oil objective N.A.=1.2). Following the comment of the reviewer we now provide additional information. A laser power less than 10% was applied for scanning in all channels. Laser power and gain were kept constant across samples within the same experiment (page 16).

A cell is defined as polarized when the ratio between the apical membrane and the cytoplasm signal intensity exceeds 1.5. A cell is defined as GATA3 positive when the nucleus to cytoplasm signal intensity exceeds 1.5. We have added these criteria in the Materials and methods section – page 17. For all images presented, we chose the mid-plane of the embryo. We will provide all our quantifications as Source Data Tables following the journal guidelines.

Statistical methods also need to be described. The nature of the tests used are not even mentioned for Figure 3 and 4 legends. Also, for example, correlations are simply given as: “correlating positively with cell numbers (p = 0.0112) (Extended Data Figure 1i)”. No correlation value is given and no detail on the test used to get the p value either.Since the statistics are not described enough, it is difficult to ascertain whether this next concern is entirely valid. Since the authors give in Figure 1c/d, Figure 2c/d or Figure 3b the number of cells measured rather than the number of embryos or of experimental replicate, I suspect that they apply their statistical tests on the number of cells pooled from embryos and replicates. This is problematic. In the case of number of polarized or GATA3 positive cells within embryos, the data need to be pooled per embryo, displayed in the figures and tested (most likely with a Student’s t test) as such. Otherwise, it is unclear, whether we see a few embryos with many polarized cells or many embryos with few polarized cells, and this can also create statistical bias with embryos having many cells weighing much more than those with fewer cells. In the case, of the percentage of polarized cells (Figure 2c/d), the data need to be pooled by replicates and tested as such (the Chi squared test compares independent replicates, which cells from the same embryo are not). In any case, the data need to be presented in the way that they are tested (i.e. not shown as pooled cells). As such, it is misleading and potentially wrong.Finally, statistical tests need to be performed for more conditions for example between control medium and DMSO as it seems to have a strong effect per se and for cell number Fig2e (see my previous point).

We thank the reviewer for this comment, as it has allowed us to strengthen our conclusions. We agree that it is better to assess the degree of polarization using embryos as the experimental unit, and not only cells. For this reason, we have now calculated the percentage of polarised cells per embryo as: number of F-actin positive cells/number of outside cells and number of Par6 positive cells/number of outside cells. Next, we have binned the data into three groups, and perform a Chi-square test. The only quantifications in which we have continue to use the cell as the experimental unit are those in which we measure fluorescence intensity levels, as the polarisation of a cell would be independent from the polarisation of other cells in the embryo (Johnson and Ziomek, 1981).

Following the suggestion of the reviewer, we have also performed statistical comparisons between the control group and the DMSO-treated groups for all the graphs.

3) The narrative of the manuscript often exaggerates some aspects, neglects others and is occasionally not based on established facts. For example, the idea that GATA3 would behave differently in mouse and human is not in agreement with the literature as suggested by the authors: “Finally, we show that, in contrast to the mouse embryo, the expression of the key extra-embryonic determinant, GATA37, is initiated in all cells and not only in extra-embryonic lineage precursors. Subsequently, the cell polarity machinery restricts the expression of GATA3 to the outer cells.” “Strikingly, GATA3 was localized to the nucleus in both polarized and non-polarized cells, although the nuclear signal intensity of GATA3 was significantly higher in polarized cells. These results suggest that cell polarization is not essential for initiation of cell fate specification in the human embryo, but rather that cell polarization positively regulates GATA3 expression (Figure 3a,c)”. The study from 2016 cited by the authors treats of human embryos and showed previously that GATA3 is initially expressed the ICM as well as in the TE. The same was found previously in the mouse as described precisely in Ralston et al. 2010. This is also consistent with another study in the mouse from Home et al. 2009, which finds Gata3 expressed as early as at the 4-cell stage indicating it is present in the precursors to both TE and ICM.

We thank the reviewer for this comment, which has helped us to make our statement clearer. We would like to point out that previous studies showed that GATA3 expression is restricted to the outer cells by the early blastocyst stage in the mouse embryo (Ralston et al., 2010; Home et al., 2009). Here, we report that GATA3 expression at the early blastocyst stage of the human embryo can be detected in both outer and inner cells, although it is expressed at a higher level in outer cells. In our opinion, this is different from what has been described in the mouse embryo. Following the reviewer’s comment, and to make our conclusion clearer, we have changed the relevant sentence in the manuscript and now state that “These results suggest that in human embryos GATA3 expression is initiated independently of embryo polarisation, but its expression and nuclear localisation is reinforced by the acquisition of apicobasal polarity” (page 5).

Another example concerns the duration of compaction in human being much longer than in mouse. Authors state: “taking an average of 13 hours to complete (in contrast to 3-4 hours in the mouse embryo6)“. I do not see in the study from Zhu et al. 2017 where the data showing that compaction in the mouse takes 3-4 hours are. On the other hand, there are other data on the duration of compaction by Fierro-Gonzalez et al. 2013 or Maître et al. 2015, or as described in the review from White et al. 2016 (reference number 9 of the present manuscript) that report that compaction in the mouse takes at least 10 h and often continues into the 16-cell stage (making it longer than 10 h).

In our previous study of the development of the mouse embryo (Zhu et al., 2017), we recorded embryo development using time-lapse microscopy. This led us to identify two phases of compaction and polarisation. Phase I covers the early 8-cell stage (1h post-division) to mid- 8-cell stage (3-4h post-division) (Figure 1a and Supplementary Figure 1d in Zhu et al., 2017). And Phase II covers mid-late 8-cell stage (5–8 h post cell division). We measured the interblastomere angle as a readout of compaction, as we did in the current study in the human embryo, and found that the inter-blastomere angle increases from less than 75 degree to more than 150 degree in Phase I (Figure 1c and Supplementary Figure 1f in Zhu et al., 2017), indicating embryo compaction. Therefore, our results suggest that the process of compaction in the mouse embryo is completed within 3-4 hours (Figure 1a,c,d and Supplementary Figure 1f (Zhu et al., 2017)). This result is in agreement with the numerous landmark studies from the 80s, which also documented that mouse embryo compaction is completed at the 8-cell stage (Levy et al., 1986, Pratt et al., 1982). We have used here the same tools to measure the time of compaction in the human embryo to be able to relate this data between the two species.

One last example about the supposed difference in timing of polarization and TE differentiation.“Surprisingly, we found that in 37% of embryos, inside cells were present with less than eight outer polarized cells (Figure 4a, n=27). This suggests that the generation of inside cells in the human embryo might not require the presence of a layer of outer polarized cells. ” Having more than 8 polarized cells does not mean that there is a layer of polarized cells at the surface of mouse embryos. This gives a wrong picture of mouse development as the layer only closes late in the 16-cell stage (Zenker et al. 2018).

We would like to clarify that we state “a layer of outer polarized cells” to indicate a situation where all cells localizing to the outside of the embryos have an apical domain. To clarify this situation, we now say that “this suggests that the generation of inside cells in the human embryo might not require the preceding establishment of polarization in all cells” (page 6).

In addition to claims in contradiction with the literature, too many statements are exaggerated.For example:“Of note, we found that although PLC inhibitor treatment significantly affected cell polarization, it did not have a significant effect on embryo compaction compared to the DMSO control group (Extended Data Figure 2d). These results indicate that compaction and polarization are independently triggered in the human embryo and that embryo polarization is regulated by PLC activity”. That’s clearly because DMSO on its own has a strong effect on compaction at the levels used by the authors.

We agree with the reviewer and for this reason we have repeated the PLC inhibitor experiments to increase the number of embryos analysed. As it can be seen in the new Figure S3, although we don’t detect a significant difference between DMSO 7.5 µM and PLCi 7.5 µM, PLC inhibition slightly decreases the percentage of compacted embryos. Therefore, we conclude that PLC does not play a major role during compaction.

In the abstract: “detailed characterization of morphological and transcriptional events”. That’s a bit much of a claim for cell counting and antibody staining against a single marker.

We have now expanded our analysis. For example, we add analyses of other trophectoderm transcription factors such as YAP1 (new Extended Data Figure 4a, b). In accord with our findings for GATA3, we find that the nucleus-to-cytoplasmic ratio of YAP1 (an indicator of its transcription activity) is higher in outer cells than in the inner cells. We also provide further analyses of polarity markers (please see below).

“The fact that expression of the TE lineage determinant is initiated independently of embryo polarization”.Concluding this strongly that it is independent of polarization based on a single polarity marker is bold.

We now provide additional analyses of polarity markers. Our new results show that aPKC displays a similar localization pattern to PARD6 (new Extended Data Figure 2). We agree that we have not analysed as many markers as we would in mouse embryos, but working with a sufficiently large number of human embryos is challenging. Here we have analysed a total of 325 human embryos, which to our knowledge is unprecedented.

“However, we also show that the onset of cell fate specification – cell allocation and expression of lineage determining genes – is independent of cell polarization.” That’s incorrect according to the authors own results on PLC inhibition which prevents GATA3 accumulation.

We agree and have now reworded our statement and say “These results suggest that in human embryos GATA3 expression is initiated independently of embryo polarisation, but its expression and nuclear localisation is reinforced by the acquisition of apicobasal polarity” (page 5).

“Interestingly, we found that the cell cycle time significantly decreased from the 4/8-cell stage transition (22.15 hours ± 0.9192) to the 8/16-cell stage transition (14.49 ± 4.189)”. This is based only on 2 embryos for the 4/8-cell stage transition as far as I could find. What test was used? What happened to those embryos afterwards? Could they be measured for their 8/16 transition as well?

Cell cycle length could only be assessed for blastomeres undergoing two serial identifiable cytokinesis during the time-lapse video. Some blastomeres did not divide twice during the video. Moreover, some divisions become obscured behind the compacting/compacted morulae so they were not traceable (because human embryos cannot be easily marked with fluorescent markers as the mouse embryos). Assessable cell cycle lengths were further stratified by their corresponding cell-stage transition. Therefore, the cell cycle length of the 2 embryos whose blastomeres divided within the 4/8 cell-stage transition could not be annotated further. Having said this, since we only have 2 embryos for the 4/8-cell stage transition we have removed the claim of significance, as suggested by the reviewer.

The final schematic in Figure 4 displays a connection between PLC and GATA3 in mouse for which there is no data, to my knowledge. If the authors want to compare mouse and human that closely, they should repeat the experiments with Gata3 and PLC inhibitors in the mouse.

We thank reviewer for the comment. In the model we wish to show that PLC-PKC regulates the apical domain, which in turn regulates GATA3 expression through Hippo signalling. We have changed the schematic to emphasize this point clearer.

“In addition, we also found that the size of the inside cells was highly variable, ranging from 229.123 to 1896.4 μm.” and “This observation implies a possibility that, as the human embryo compact, having on average more cells than that in the mouse, the geometrical constriction generated by compaction may allow some cells to acquire an inner position without asymmetric cell divisions. ” This measurement is poorly described and based on the data reported in Figure 4d (which has no unit by the way) is anecdotal at best. How many of the outer cells are that large? Why specifically asymmetric division and division in general?

We apologize for the lack of detail and now provide a full description of how the quantification was done (please see the revised Methods section, page 17). The reviewer is also asking how many of the outer cells are large. This is shown in Figure 4d. The histogram represents the number of embryos in each bin (each bin corresponds a specific cell size). The 8-cell stage blastomere with the smallest size measures 895.432 µm^2^. Of the 53 inside blastomeres analysed, 18 (34%) are bigger than 895.432 µm^2^. We have specified this in the description of the results.

We very much thank the reviewers for their time and the many very valuable comments that helped us to improve our manuscript.

References

Α Scientists In Reproductive, M. and Embryology, E. S. I. G. O. 2011. The Istanbul consensus workshop on embryo assessment: proceedings of an expert meeting. *Hum Reprod,* 26**,** 1270-83.

Cimadomo, D., Fabozzi, G., Vaiarelli, A., Ubaldi, N., Ubaldi, F. M. and Rienzi, L. 2018. Impact of Maternal Age on Oocyte and Embryo Competence. *Front Endocrinol (Lausanne),* 9**,** 327.

Johnson, M. H. and Ziomek, C. A. 1981. Induction of polarity in mouse 8-cell blastomeres: specificity, geometry, and stability. *J Cell Biol,* 91**,** 303-8.

Levy, J. B., Johnson, M. H., Goodall, H. and Maro, B. 1986. The timing of compaction:

control of a major developmental transition in mouse early embryogenesis. *J Embryol Exp Morphol,* 95**,** 213-37.

Petropoulos, S., Edsgard, D., Reinius, B., Deng, Q., Panula, S. P., Codeluppi, S., Plaza Reyes, A., Linnarsson, S., Sandberg, R. and Lanner, F. 2016. Single-Cell RNA-Seq Reveals Lineage and X Chromosome Dynamics in Human Preimplantation Embryos. *Cell,* 165**,** 1012-26.

Pratt, H. P., Ziomek, C. A., Reeve, W. J. and Johnson, M. H. 1982. Compaction of the mouse embryo: an analysis of its components. *J Embryol Exp Morphol,* 70**,** 113-32.

Reig, A., Franasiak, J., Scott, R. T., Jr. and Seli, E. 2020. The impact of age beyond ploidy:

outcome data from 8175 euploid single embryo transfers. *J Assist Reprod Genet,* 37**,** 595-602.

Zhu, M., Leung, C. Y., Shahbazi, M. N. and Zernicka-Goetz, M. 2017. Actomyosin polarisation through PLC-PKC triggers symmetry breaking of the mouse embryo. *Nat Commun,* 8**,** 921.

Responses to 2^nd^ round of reviewers comments:

Referee #1 (Remarks to the Author):My concerns have been addressed in the revised version of the manuscript. Only remaining issue is that the concentration of DMSO used in vehicle control treatments is not indicated. Maybe the authors should provide the inhibitor stock concentration, so that the amount of DMSO used in the medium as control can be easily quantified.

We thank the reviewer for their critical evaluation of the manuscript and we apologise for the omission. The DMSO concentration is 35 mM (vehicle control of experimental group 5 µM PLC inhibitor) and 52 mM (vehicle control of experimental group 7.5 µM PLC inhibitor). We are grateful for the referee’s support. We would like to mention that the DMSO concentration is not mentioned in Gerri et al.

Referee #3 (Remarks to the Author):I thank very much the authors from the study “Mechanism of cell polarisation and first lineage segregation in the human embryo” for answering my concerns and improving their manuscript. Despite these careful answers and improvements, I am not convinced by the data and explanations provided by the authors. My initial concerns remain. The manuscript contains several statements that are in disagreement with the authors own data or with the literature. The data are often shown in a way that could easily mislead the readers. Some key methodological details are still missing. Most importantly, I suspect a major flaw in experimental design, which makes the data difficult to publish in this journal or any other. In the end, the study broadly confirms what is known from the mouse and I am surprised and concerned by the large number of human embryos that were used for answering such a simple (and nevertheless essential) question.

We thank the reviewer for his/her comments and provide a detailed response to the criticism below.

Major issues:1)The effects of DMSO that were reported in the first version prevent from supporting the conclusions of the authors. Importantly, the authors still do not indicate the concentration of DMSO that is used despite both Referee 1 and myself deeming this information essential. In case this was not clear to the authors, the “5 µM DMSO” label does not indicate the DMSO concentration would be 5 µM (I hope).

We apologise for the omission. The concentration of DMSO that was used is 35 mM (vehicle control of experimental group 5 µM PLC inhibitor) and 52 mM (vehicle control of experimental group 7.5 µM PLC inhibitor). We have specified the different vehicle groups as 5 µM and 7.5 µM so the correspondence with the experimental PLC inhibitor treated groups is clear. But we will indicate the specific DMSO concentration in the methods section. We would like to mention that the DMSO concentration is not mentioned in Kathy’s paper just published in Nature (Gerri et al.)

The authors performed additional experiments to check for the effects of DMSO on embryo development (FS3d). Between the successive versions of FigS3d, control embryos went from 24 compacted to 40 compacted, 21 non-compacted and 12 dead/arrested. It seems that the group of control embryos that were added to the revised version performed particularly poorly. DMSO went from 15 compacted, 19 non-compacted and 6 dead/arrested to DMSO 5 µM 16/12/3 and DMSO 7.5 µM 10/13/4. It seems that the group of DMSO embryos that were added performed particularly well. Can the authors explain the reasons for the discrepancy between the initial control and DMSO experiments and the ones of the revised manuscript? It cannot be because of sample size as claimed by the authors in their rebuttal letter since the sample size is not dramatically different.

We thank the reviewer for this comment and are happy to clarify the new result. We have experienced a big variation in the developmental dynamics of human embryos depending on the experiment. We believe this is in part due to patient-specific differences (in each experiment, embryos are used from several patients). As the reviewer points out, in the last experiment a large proportion of control untreated embryos did not compact. Following the suggestions of the reviewers, we split the DMSO group into two, to reflect the different concentrations of DMSO used. For this reason, throughout the manuscript all graphs display two vehicle groups. We will make it clear in the revised manuscript. Having said this, we now provide all the raw data: time-lapse videos and the associated quantifications. Our manuscript has allowed us to compare the development of untreated embryos with DMSO-treated (vehicle) embryos. This comparison is missing in Gerry et al., and we believe it provides an extra layer of control to our study.

Importantly, it seems that no additional experiments with 5 µM U73122 were performed (sample size remained the same). Nevertheless, the first batch of experiments with 5 µM U73122 are compared to the new batch of control. It seems incorrect to compare experiments that were not done concomitantly and especially when the batches of embryos behave so differently.

We did not repeat the 5 µM group since the biggest differences were obtained with 7.5 µM. As the reviewer points out, we have used quite a large number of embryos and therefore, we decided to focus on the most relevant conditions, and pool embryos from all independent experiments into a single graph. Having said this, if the reviewer believes it would be more appropriate to separate the control group into two, depending on whether embryos were thawed and cultured together with the 5 µM or 7.5 µM conditions, we would be happy to do it. We can also point this out explicitly in the paper.

The addition of siRNA could be useful but does not present a good control of the effective KD. Therefore it only marginally raises the confidence in the proposed conclusions.We would like to emphasise that this is the first time that siRNAs have been used to study gene function in human embryos (to our knowledge). This is an extremely challenging experiment that involves 4 different labs in 4 different countries. During the pandemic we managed to validate the knock-down of both PLC isoforms in human embryos as shown in the revised extended Figure 3f. Injection of PLC siRNAs significant downregulates PLCE1 and PLCB1 mRNA levels by 90% and 70% respectively.Therefore, the experiment at the core of the study is flawed from its inception and the additional experiments provided give me little confidence on the ability of the authors to fix this flaw.

We apologize for the confusion, but we are not aware of any specific flaw in our experimental design. Our experimental design for the inhibitor treatments includes all the appropriate controls (untreated embryos, vehicle control embryos at two different concentrations, and experimental groups with two different inhibitor concentrations). All of this is done on human embryos and this is extremely difficult to perform so many experiments on human embryos and an enormous amount of effort and care went to those experiments. Our experimental design for the siRNA treatments includes a scramble control and PLC siRNA experimental group. We have validated the siRNAs in revised extended Figure 3f. Injection of PLC siRNAs significantly downregulates PLCE1 and PLCB1 mRNA level by 90% and 70% respectively. We have analysed embryos across various independent experiments and from multiple couples due to the variability of IVF human embryos.

2) The conclusions from the authors are often not supported by the data provided or by the literature. The abstract claims that PLC triggers actin polarization but this seems fine in Figure 2B and 2c (only 7.5µM shows a statistically significant effect and the DMSO control seems to behave differently from control and 5 µM DMSO embryos, but is not tested, even though I suggested the authors to provide all tests in the initial review and the authors claiming to have done so).

PLC triggers actin polarization as can be seen in Figure 2b and 2c. We have provided the relevant statistical comparison, in this case 7.5 µM DMSO vs. 7.5 µM PLC inhibitor. Adding the additional statistical comparisons that the reviewer requests is straightforward: control vs. 7.5 µM PLC inhibitor, Chi-square test, *p=0.0361; 5 µM DMSO vs. 7.5 µM PLC inhibitor, Chi-square test, ****p<0.0001. We apologise for not providing this straight away.

The abstract claims that GATA3 is found in all cells, when this is clearly not the case (see Figure 3A).

We will correct the abstract to explain this result better.

The abstract claims that polarity enhances the expression of GATA3 but this is based on a very small difference in immunostaining intensity (Fig3e), which is a method with poor resolution and important potential experimental artifacts (light scattering, antibody binding, …).

We respectfully disagree with the referee as the analysis of protein expression by immunofluorescence is an accepted standard in developmental biology. Quantifications of immunofluorescence images in mammalian embryos have been performed for the past 20 years not only by our group but by many others (for example Xenopoulos et al., Cell Reports, 2015; White et al., Cell, 2016; Posfai et al., *eLife*, 2017; Maître et al., Nature, 2016; Chan et al., Nature, 2019; Lim et al., Nature, 2020 and many other papers). The difference in GATA3 levels between control and PLC inhibitor treated embryos is small, but clearly significant. As discussed in the manuscript this supports a model of GATA3 expression been initiated independently of polarization but reinforced by it. We would like to emphasise that Gerry et al. uses immunofluorescence data to quantify the levels of GATA3 in single cells. The differences they observe between control and experimental groups are similar to the differences we report.

The duration of compaction is claimed to be longer in human (> half day) than in mouse (< hald fay) (Fig4e) despite the recent literature I mentioned in the previous review and the own measurements of the authors given in FigS1j with about half human embryos compacting in less than 6 h.

The short duration of compaction time in the mouse has been documented by many studies (for example, around 4.2–5.2 h post division by Ziomek and Johnson, Cell, 1980; 6h by Maro and Pickering, J. Embryol. Exp. Morphol., 1984; 3-5h by Pratt et al., J. Embryol. Exp. Morphol., 1982 and our own study in the mouse embryo 3-4h by Zhu et al., Nat Comm, 2017). In our analysis in Figure S1j, we found that the human embryo compaction time is highly variable (range from 2-22h) but on average takes a longer time than in the mouse (7h in human). Therefore, we concluded that in human embryo the compaction process takes longer than in the mouse. To make the conclusion more concise, we will reword the conclusion by saying “duration of compaction in human embryo is highly variable and takes a longer average time than in mouse.”

The review article mentioned in the previous review does not provide a specific timing of compaction. Moreover, the authors of this review have broadened the compaction concept to include cell fate decisions happening in the 16-cell stage. As our work refers to specifically the compaction process per se (which in the mouse is completed at the 8-cell stage) we are not sure how relevant this review citation can be.

The additional timelapse of human embryos provided to track cell internalization is claimed to provide evidence that a cell internalizes “independently of polarity and cell divisions” (page 6). However, the FigS4 clearly shows an asymmetric division.

In the example shown in Figure S4 a division took place prior to the initiation of compaction (the mother cell is labelled in yellow and two daughter cells are labelled in cyan and red). One of the two daughter cells (labelled in red) was initially localised slightly inside of the embryo relative to the other cells. Afterwards, the compaction process rearranged cell positions by sealing all cell boundaries, and as a result this cell became localised entirely on the inside of the embryo. In the mouse embryo, asymmetric divisions take place once compaction and polarization have been completed. Therefore, there are no asymmetric divisions taking place in the example provided.

3) The presentation of data can be misleading. This remains unchanged from the previous comments I made and to the ones of the other referees. For example, giving the cell number instead of the embryo number in Figure 1C-f makes it impossible to know whether a few embryos show more polarized cells or whether many embryos show a few polarized cells.

We took into consideration the initial comment of the reviewer and modified the key graphs in the revised version (e.g. Figure 2c and 2d). We can also modify the graphs shown in Figure 1 for consistency with the rest of the manuscript. The only graphs in which we would like to keep the cell-level analysis are those in which fluorescence intensity is measured, as the apical enrichment of polarity proteins will be heterogenous between cells of the same embryo. Since the reviewer is concerned about the inter- and intra-embryo variability, we could present the analysis indicating to which embryo each cell belongs. Having said this, we would like to mention that Gerry et al. also analyse immunofluorescence images at the cell level.

We strongly believe that this is extremely valuable work as it defines for the first time those steps upstream of cell polarization in the human embryo. We apologise to the referee for failing to take care of some points in the revised manuscript. These are now all corrected. We very much hope that both the editor and the two referees realise that the multiple groups working on this paper did not make mistakes in the design and execution of these experiments and that all the conclusions of the paper are important and valid.